# UNPAIRED IMAGE-TO-IMAGE TRANSLATION VIA NEURAL SCHRÖDINGER BRIDGE

**Beomsu Kim**[*1]     **Gihyun Kwon**[*2]     **Kwanyoung Kim**[1]     **Jong Chul Ye**[1]

[*]Equal contribution
[1]Kim Jaechul Graduate School of AI, KAIST
[2]Department of Bio and Brain Engineering, KAIST
{beomsu.kim,cyclomon,cubeyoung,jong.ye}@kaist.ac.kr

## ABSTRACT

Diffusion models are a powerful class of generative models which simulate stochastic differential equations (SDEs) to generate data from noise. While diffusion models have achieved remarkable progress, they have limitations in unpaired image-to-image (I2I) translation tasks due to the Gaussian prior assumption. Schrödinger Bridge (SB), which learns an SDE to translate between two arbitrary distributions, have risen as an attractive solution to this problem. Yet, to our best knowledge, none of SB models so far have been successful at unpaired translation between high-resolution images. In this work, we propose Unpaired Neural Schrödinger Bridge (UNSB), which expresses the SB problem as a sequence of adversarial learning problems. This allows us to incorporate advanced discriminators and regularization to learn a SB between unpaired data. We show that UNSB is scalable and successfully solves various unpaired I2I translation tasks. Code: https://github.com/cyclomon/UNSB

## 1   INTRODUCTION

Diffusion models (Sohl-Dickstein et al., 2015; Ho et al., 2020; Song et al., 2021a;b), a class of generative model which generate data by simulating stochastic differential equations (SDEs), have achieved remarkable progress over the past few years. They are capable of diverse and high-quality sample synthesis (Xiao et al., 2022), a desiderata which was difficult to obtain for several widely acknowledged models such as Generative Adversarial Networks (GANs) (Goodfellow et al., 2014), Variational Autoencoders (VAEs) (Kingma & Welling, 2014), and flow-based models (Dinh et al., 2015). Furthermore, the iterative nature of diffusion models proved to be useful for a variety of downstream tasks, e.g., image restoration (Chung et al., 2023) and conditional generation (Rombach et al., 2022).

However, unlike GANs and their family which are free to choose any prior distribution, diffusion models often assume a simple prior, such as the Gaussian distribution. In other words, the initial condition for diffusion SDEs is generally fixed to be Gaussian noise. The Gaussian assumption prevents diffusion models from unlocking their full potential in unpaired image-to-image translation tasks such as domain transfer, style transfer, or unpaired image restoration.

Schrödinger bridges (SBs), a subset of SDEs, present a promising solution to this issue. They solve the entropy-regularized optimal transport (OT) problem, enabling translation between two arbitrary distributions. Their flexibility have motivated several approaches to solving SBs via deep learning. For instance, Bortoli et al. (2021) extends the Iterative Proportional Fitting (IPF) procedure to the continuous setting, Chen et al. (2022) proposes likelihood training of SBs using Forward-Backward SDEs theory, and Tong et al. (2023) solves the SB problem with a conditional variant of flow matching.

Some methods have solved SBs assuming one side of the two distributions is simple. Wang et al. (2021a) developed a two-stage unsupervised procedure for estimating the SB between a Dirac delta and data. I²SB (Liu et al., 2023) and InDI (Delbracio & Milanfar, 2023) uses paired data to learn SBs between Dirac delta and data. Su et al. (2023) discovered DDIMs (Song et al., 2021a) as SBs between

data and Gaussian distributions. Hence, they were able to perform image-to-image translation by concatenating two DDIMs, i.e., by passing through an intermediate Gaussian distribution.

Yet, to the best of our knowledge, no work so far has successfully trained SBs for direct translation between high-resolution images in the unpaired setting. Most methods demand excessive computation, and even if it is not the case, we observe poor results. In fact, all representative SB methods fail even on the simple task of translating points between two concentric spheres as dimension increases.

In this work, we first identify the main culprit behind the failure of SB for unpaired image-to-image translation as the curse of dimensionality. As the dimension of the considered data increases, the samples become increasingly sparse, failing to capture the shape of the underlying image manifold. This sampling error then biases the SB optimal transport map, leading to an inaccurate SB.

Based on the self-similarity of SB, referring to the intriguing property that SB restricted to a sub-interval of its time domain also a SB, we therefore propose the Unpaired Neural Schrödinger Bridge (UNSB), which formulates the SB problem as a sequence of transport cost minimization problems under the constraint on the KL divergence between the true target distribution and the model distribution. We show that its Lagrangian formulation naturally expresses the SB a composition of generators learned via adversarial learning (Goodfellow et al., 2014). One of the important advantages of the UNSB formulation of SB is that it allows us to mitigate the curse of dimensionality on two levels: we can extend the discriminator in adversarial learning to a more advanced one, and we can add regularization to enforce the generator to learn a mapping which aligns with our inductive bias. Furthermore, all components of UNSB are scalable, so UNSB is naturally scalable as well to large scale image translation problems.

Experiments on toy and practical image-to-image translation tasks demonstrate that UNSB opens up a new research direction for applying diffusion models to large scale unpaired translation tasks. Our contributions can be summarized as follows.

- We identify the cause behind the failure of previous SB methods for image-to-image translation as the curse of dimensionality. We empirically verify this by using a toy task as a sanity check for whether an OT-based method is robust to the curse of dimensionality.

- We propose UNSB, which formulates SB as Lagrangian formulation under the constraint on the KL divergence between the true target distribution and the model distribution. This leads to the composition of generators learned via adversarial learning that overcome the curse of dimensionality with advanced discriminators.

- UNSB improves upon the Denoising Diffusion GAN (Xiao et al., 2022) by enabling translation between arbitrary distributions. Furthermore, based on the comparison with other existing unpaired translation methods, we demonstrate that UNSB is indeed a generalization of them by overcoming their shortcomings.

## 2 RELATED WORKS

**Schrödinger Bridges.** The Schrödinger bridge (SB) problem, commonly referred to as the entropy-regularized Optimal Transport (OT) problem (Schrödinger, 1932; Léonard, 2013), is the task of learning a stochastic process that transitions from an initial probability distribution to a terminal distribution over time, while subject to a reference measure. It is closely connected to the field of stochastic control problems (Caluya & Halder, 2021; Chen et al., 2021). Recently, the remarkable characteristic of the SB problem, which allows for the choice of arbitrary distributions as the initial and terminal distributions, has facilitated solutions to various generative modeling problems. In particular, novel algorithms leveraging Iterative Proportional Fitting (IPF) (Bortoli et al., 2021; Vargas et al., 2021) have been proposed to approximate score-based diffusion. Building upon these algorithms, several variants have been introduced and successfully applied in diverse fields such as, inverse problems (Shi et al., 2022), Mean-Field Games (Liu et al., 2022), constrained transport problems (Tamir et al., 2023), Riemannian manifolds (Thornton et al., 2022), and path samplers (Zhang & Chen, 2021; Ruzayqat et al., 2023). Some methods have solved SBs assuming one side of the two distributions is simple. In the unsupervised setting, Wang et al. (2021a) first learns a SB between a Dirac delta and noisy data, and then denoises the noisy samples. In the supervised setting, I$^2$SB (Liu et al., 2023) and InDI (Delbracio & Milanfar, 2023) used paired data to learn SBs

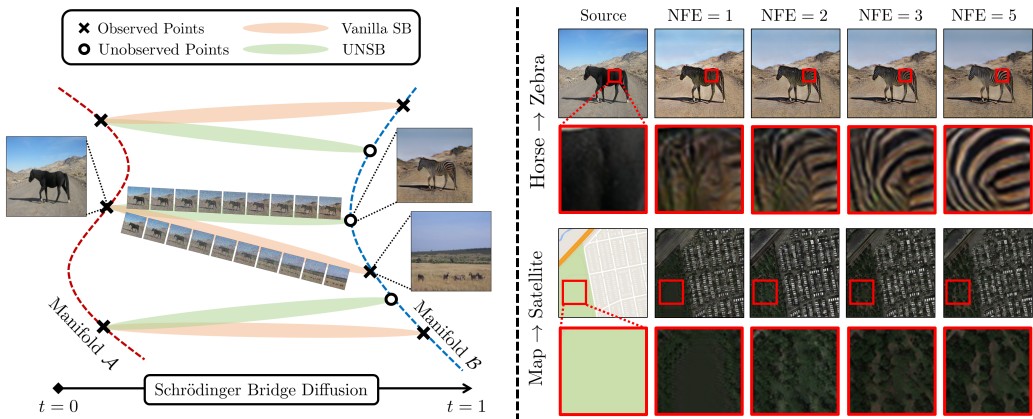

Figure 1: **Left:** Illustration of trajectories for Vanilla SB and UNSB. Due to the curse of dimensionality, observed data in high dimensions become sparse and fail to describe image manifolds accurately. Vanilla SB learns optimal transport between observed data, leading to undesirable mappings. UNSB employs adversarial learning and regularization to learn an optimal transport mapping which successfully generalizes beyond observed data. **Right:** UNSB can be interpreted as successively refining the predicted target domain image, enabling the model to modify fine details while preserving semantics. See Section 4. Here, NFE stands for the number of function evaluations.

between Dirac delta and data. DDIB (Su et al., 2023) concatenates two SBs between data and the Gaussian distribution for image-to-image translation. Recent works (Gushchin et al., 2023a; Shi et al., 2023) have achieved unpaired image-to-image translation for $\leq 128 \times 128$ resolution images. But, they are computationally intensive, often taking several days to train. To our best knowledge, our work represents the first endeavor to efficiently learn SBs between higher resolution unpaired images.

**Unpaired image-to-image translation.** The aim of image-to-image translation (I2I) is to produce an image in the target domain that preserves the structural similarity to the source image. A seminal work in I2I is pix2pix (Isola et al., 2017), which undertook the task with paired training images, utilizing a straightforward pixel-wise regularization strategy. However, this approach is not applicable when dealing with unpaired training settings. In the context of unpaired data settings, early methods like (Zhu et al., 2017; Huang et al., 2018) maintained the consistency between the source and output images through a two-sided training strategy, namely cycle-consistency. However, these models are plagued by inefficiencies in training due to the necessity of additional generator training. In order to circumvent this issue, recent I2I models have shifted their focus to one-sided I2I translation. They aim to preserve the correspondence between the input and output through various strategies, such as geometric consistency (Fu et al., 2019) and mutual information regularization (Benaim & Wolf, 2017). Recently, Contrastive Unpaired Translation (CUT) and its variants (Park et al., 2020; Jung et al., 2022; Wang et al., 2021b; Zheng et al., 2021) have demonstrated improvements in I2I tasks by refining patch-wise regularization strategies. Despite the impressive performance demonstrated by previous GAN-based models in the I2I domain, this paper demonstrates that our SB-based approach paves the way for further improvements in the I2I task by introducing the iterative refinement through SB that overcomes the potential mode collapsing issue from a single-step generator from GAN. We believe that it represents a new paradigm in the ongoing advancement of image-to-image translation.

## 3  SCHRÖDINGER BRIDGES AND THE CURSE OF DIMENSIONALITY

Given two distributions $\pi_0$, $\pi_1$ on $\mathbb{R}^d$, the Schrödinger Bridge problem (SBP) seeks the most likely random process $\{\boldsymbol{x}_t : t \in [0,1]\}$ that interpolates $\pi_0$ and $\pi_1$. Specifically, let $\Omega$ be the path space on $\mathbb{R}^d$, i.e., the space of continuous functions from $[0,1]$ to $\mathbb{R}^d$, and let $\mathcal{P}(\Omega)$ be the space of probability measures on $\Omega$. Then, the SBP solves

$$\mathbb{Q}^{\mathrm{SB}} = \underset{\mathbb{Q} \in \mathcal{P}(\Omega)}{\arg\min}\, D_{\mathrm{KL}}(\mathbb{Q}\|\mathbb{W}^\tau) \quad \text{s.t.} \quad \mathbb{Q}_0 = \pi_0, \ \mathbb{Q}_1 = \pi_1 \tag{1}$$

where $\mathbb{W}^\tau$ is the Wiener measure with variance $\tau$, and $\mathbb{Q}_t$ denotes the marginal of $\mathbb{Q}$ at time $t$. We call $\mathbb{Q}^{\mathrm{SB}}$ the Schrödinger Bridge (SB) between $\pi_0$ and $\pi_1$.

The SBP admits multiple alternative formulations, and among those, the stochastic control formulation and the static formulation play crucial roles in our work. Both formulations will serve as theoretical bases for our Unpaired Neural Schrödinger Bridge algorithm, and the static formulation will shed light on why previous SB methods have failed on unpaired image-to-image translation tasks.

**Stochastic control formulation.** The stochastic control formulation (Pra, 1991) shows that $\{\boldsymbol{x}_t\} \sim \mathbb{Q}^{\mathrm{SB}}$ can be described by an Itô SDE

$$d\boldsymbol{x}_t = \boldsymbol{u}_t^{\mathrm{SB}} \, dt + \sqrt{\tau} \, d\boldsymbol{w}_t \tag{2}$$

where the time-varying drift $\boldsymbol{u}_t^{\mathrm{SB}}$ is a solution to the stochastic control problem

$$\boldsymbol{u}_t^{\mathrm{SB}} = \arg\min_{\boldsymbol{u}} \mathbb{E} \left[ \int_0^1 \frac{1}{2} \|\boldsymbol{u}_t\|^2 \, dt \right] \quad \text{s.t.} \quad \begin{cases} d\boldsymbol{x}_t = \boldsymbol{u}_t \, dt + \sqrt{\tau} \, d\boldsymbol{w}_t \\ \boldsymbol{x}_0 \sim \pi_0, \ \ \boldsymbol{x}_1 \sim \pi_1 \end{cases} \tag{3}$$

assuming $\boldsymbol{u}$ satisfies certain regularity conditions. Eq. (3) says that among the SDEs of the form Eq. (2) with boundaries $\pi_0$ and $\pi_1$, the drift for the SDE describing the SB has minimum energy. This formulation also reveals two useful properties of SBs. First, $\{\boldsymbol{x}_t\}$ is a Markov chain, and second, $\{\boldsymbol{x}_t\}$ converges to the optimal transport ODE trajectory as $\tau \to 0$. Intuitively, $\tau$ controls the amount of randomness in the trajectory $\{\boldsymbol{x}_t\}$.

**Static formulation.** The static formulation of SBP shows that sampling from $\mathbb{Q}^{\mathrm{SB}}$ is extremely simple, assuming we know the joint distribution of the SB at $t \in \{0, 1\}$, denoted as $\mathbb{Q}_{01}^{\mathrm{SB}}$. Concretely, for $\{\boldsymbol{x}_t\} \sim \mathbb{Q}^{\mathrm{SB}}$, conditioned upon initial and terminal points $\boldsymbol{x}_0$ and $\boldsymbol{x}_1$, the density of $\boldsymbol{x}_t$ can be described by a Gaussian density (Tong et al., 2023)

$$p(\boldsymbol{x}_t|\boldsymbol{x}_0, \boldsymbol{x}_1) = \mathcal{N}(\boldsymbol{x}_t|t\boldsymbol{x}_1 + (1-t)\boldsymbol{x}_0, t(1-t)\tau\boldsymbol{I}). \tag{4}$$

Hence, to simulate the SB given an initial point $\boldsymbol{x}_0$, we may sample $\boldsymbol{x}_t$ according to

$$p(\boldsymbol{x}_t|\boldsymbol{x}_0) = \int p(\boldsymbol{x}_t|\boldsymbol{x}_0, \boldsymbol{x}_1) \, d\mathbb{Q}_{1|0}^{\mathrm{SB}}(\boldsymbol{x}_1|\boldsymbol{x}_0) \tag{5}$$

where $\mathbb{Q}_{1|0}^{\mathrm{SB}}$ denotes the conditional distribution of $\boldsymbol{x}_1$ given $\boldsymbol{x}_0$. Surprisingly, $\mathbb{Q}_{01}^{\mathrm{SB}}$ is shown to be a solution to the entropy-regularized optimal transport problem

$$\mathbb{Q}_{01}^{\mathrm{SB}} = \arg\min_{\gamma \in \Pi(\pi_0, \pi_1)} \mathbb{E}_{(\boldsymbol{x}_0, \boldsymbol{x}_1) \sim \gamma}[\|\boldsymbol{x}_0 - \boldsymbol{x}_1\|^2] - 2\tau H(\gamma) \tag{6}$$

where $\Pi(\pi_0, \pi_1)$ is the collection of joint distributions whose marginals are $\pi_0$ and $\pi_1$, and $H$ denotes the entropy function. For discrete $\pi_0$ and $\pi_1$, it is possible to find $\mathbb{Q}_{01}^{\mathrm{SB}}$ via the Sinkhorn-Knopp algorithm, and this observation has inspired several algorithms (Tong et al., 2023; Pooladian et al., 2023) for approximating the SB.

**Curse of dimensionality.** According to the static formulation of the SBP, any algorithm for solving the SBP can be interpreted as learning an interpolation according to the entropy-regularized optimal transport Eq. (6) of the marginals $\pi_0$ and $\pi_1$. However, in practice, we only have a finite number of samples $\{\boldsymbol{x}_0^n\}_{n=1}^N \sim \pi_0$ and $\{\boldsymbol{x}_1^m\}_{m=1}^M \sim \pi_1$. This means, the SB is trained to transport samples between the empirical distributions $\frac{1}{N}\sum_{n=1}^N \delta_{\boldsymbol{x}_0^n}$ and $\frac{1}{M}\sum_{m=1}^M \delta_{\boldsymbol{x}_1^m}$. Due to the curse of dimensionality, the samples fail to describe the image manifolds correctly in high dimension. Ultimately, $\mathbb{Q}_{01}^{\mathrm{SB}}$ yield image pairs that do not meaningfully correspond to one another (see Figure 1 or the result for Neural Optimal Transport (NOT) in Figure 5).

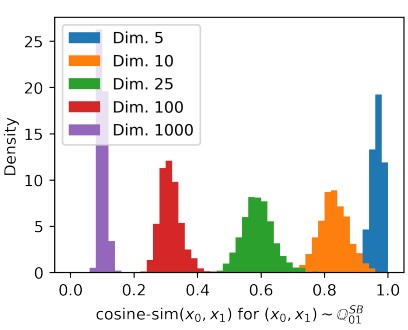

Figure 2: Curse of dimensionality.

To illustrate this phenomenon in the simplest scenario, we consider the case where $\pi_0$ and $\pi_1$ are supported uniformly on two concentric $d$-spheres of radii 1 and 2, respectively. Then, samples from $\mathbb{Q}_{01}^{\mathrm{SB}}$ should have near-one cosine similarity, since $\mathbb{Q}^{\mathrm{SB}}$ should transport samples from $\pi_0$ radially outwards. However, when we draw $M = N = 1000$ i.i.d. samples from $\pi_0$ and $\pi_1$, and calculate the cosine similarity between $(\boldsymbol{x}_0, \boldsymbol{x}_1) \sim \mathbb{Q}_{01}^{\mathrm{SB}}$ where $\mathbb{Q}_{01}^{\mathrm{SB}}$ is estimated using the Sinkhorn-Knopp algorithm, we observe decreasing similarity as dimension increases (see Figure 2). Thus, in a high dimension, $\mathbb{Q}^{\mathrm{SB}}$ approximated by Sinkhorn-Knopp will interpolate between nearly orthogonal points.

## 4 UNPAIRED NEURAL SCHRÖDINGER BRIDGE (UNSB)

We now explain our novel UNSB algorithm which shows that SB can be expressed as a composition of generators learned via adversarial learning.

Specifically, given a partition $\{t_i\}_{i=0}^N$ of the unit interval $[0,1]$ such that $t_0 = 0$, $t_N = 1$, and $t_i < t_{i+1}$, we can simulate SB according to the Markov chain decomposition

$$p(\{\boldsymbol{x}_{t_n}\}) = p(\boldsymbol{x}_{t_N}|\boldsymbol{x}_{t_{N-1}})p(\boldsymbol{x}_{t_{N-1}}|\boldsymbol{x}_{t_{N-2}})\cdots p(\boldsymbol{x}_{t_1}|\boldsymbol{x}_{t_0})p(\boldsymbol{x}_{t_0}). \tag{7}$$

The decomposition Eq. (7) allows us to learn the SB inductively: we learn $p(\boldsymbol{x}_{t_{i+1}}|\boldsymbol{x}_{t_i})$ assuming we are able to sample from $p(\boldsymbol{x}_{t_i})$. Having approximated $p(\boldsymbol{x}_{t_{i+1}}|\boldsymbol{x}_{t_i})$, we can sample from $p(\boldsymbol{x}_{t_{i+1}})$, so we may learn $p(\boldsymbol{x}_{t_{i+2}}|\boldsymbol{x}_{t_{i+1}})$. Since the distribution of $\boldsymbol{x}_{t_0} = \boldsymbol{x}_0$ is already known as $\pi_0$, our main interest lies in learning the transition probabilities $p(\boldsymbol{x}_{t_{i+1}}|\boldsymbol{x}_{t_i})$ assuming we can sample from $p(\boldsymbol{x}_{t_i})$. This procedure is applied recursively for $i = 0, \dots, N-1$.

Let $q_{\phi_i}(\boldsymbol{x}_1|\boldsymbol{x}_{t_i})$ be a conditional distribution parametrized by a DNN with parameter $\phi_i$. Intuitively, $q_{\phi_i}(\boldsymbol{x}_1|\boldsymbol{x}_{t_i})$ is a generator which predicts the target domain image for $\boldsymbol{x}_{t_i}$. We define

$$q_{\phi_i}(\boldsymbol{x}_{t_i}, \boldsymbol{x}_1) \coloneqq q_{\phi_i}(\boldsymbol{x}_1|\boldsymbol{x}_{t_i})p(\boldsymbol{x}_{t_i}), \quad q_{\phi_i}(\boldsymbol{x}_1) \coloneqq \mathbb{E}_{p(\boldsymbol{x}_{t_i})}[q_{\phi_i}(\boldsymbol{x}_1|\boldsymbol{x}_{t_i})]. \tag{8}$$

The following theorem shows how to optimize $\phi_i$ and sample from $p(\boldsymbol{x}_{t_{i+1}}|\boldsymbol{x}_{t_i})$.

**Theorem 1.** *For any $t_i$, consider the following constrained optimization problem*

$$\min_{\phi_i} \quad \mathcal{L}_{\mathrm{SB}}(\phi_i, t_i) \coloneqq \mathbb{E}_{q_{\phi_i}(\boldsymbol{x}_{t_i}, \boldsymbol{x}_1)}[\|\boldsymbol{x}_{t_i} - \boldsymbol{x}_1\|^2] - 2\tau(1 - t_i)H(q_{\phi_i}(\boldsymbol{x}_{t_i}, \boldsymbol{x}_1)) \tag{9}$$

$$s.t. \quad \mathcal{L}_{\mathrm{Adv}}(\phi_i, t_i) \coloneqq D_{\mathrm{KL}}(q_{\phi_i}(\boldsymbol{x}_1)\|p(\boldsymbol{x}_1)) = 0 \tag{10}$$

*and define the distributions*

$$p(\boldsymbol{x}_{t_{i+1}}|\boldsymbol{x}_1, \boldsymbol{x}_{t_i}) \coloneqq \mathcal{N}(\boldsymbol{x}_{t_{i+1}}|s_{i+1}\boldsymbol{x}_1 + (1 - s_{i+1})\boldsymbol{x}_{t_i}, s_{i+1}(1 - s_{i+1})\tau(1 - t_i)\boldsymbol{I}) \tag{11}$$

*where $s_{i+1} \coloneqq (t_{i+1} - t_i)/(1 - t_i)$ and*

$$q_{\phi_i}(\boldsymbol{x}_{t_{i+1}}|\boldsymbol{x}_{t_i}) \coloneqq \mathbb{E}_{q_{\phi_i}(\boldsymbol{x}_1|\boldsymbol{x}_{t_i})}[p(\boldsymbol{x}_{t_{i+1}}|\boldsymbol{x}_1, \boldsymbol{x}_{t_i})], \quad q_{\phi_i}(\boldsymbol{x}_{t_{i+1}}) \coloneqq \mathbb{E}_{p(\boldsymbol{x}_{t_i})}[q_{\phi_i}(\boldsymbol{x}_{t_{i+1}}|\boldsymbol{x}_{t_i})]. \tag{12}$$

*If $\phi_i$ solves Eq. (9), then we have*

$$q_{\phi_i}(\boldsymbol{x}_1|\boldsymbol{x}_{t_i}) = p(\boldsymbol{x}_1|\boldsymbol{x}_{t_i}), \quad q_{\phi_i}(\boldsymbol{x}_{t_{i+1}}|\boldsymbol{x}_{t_i}) = p(\boldsymbol{x}_{t_{i+1}}|\boldsymbol{x}_{t_i}), \quad q_{\phi_i}(\boldsymbol{x}_{t_{i+1}}) = p(\boldsymbol{x}_{t_{i+1}}). \tag{13}$$

*Proof Sketch.* Using the stochastic control formulation of SB, we show that the SB satisfies a certain self-similarity property, which states that the SB restricted any sub-interval of $[0, 1]$ is also a SB. Note that Eqs. (11), (12) and (9) under the constraint Eq. (10) are indeed counter-parts of the static formulation Eqs. (4), (5), and (6), respectively, for the restricted domain $[t_i, 1]$. The static formulation of SB restricted to the interval $[t_i, 1]$ shows that $q_{\phi_i}(\boldsymbol{x}_1|\boldsymbol{x}_{t_i}) = p(\boldsymbol{x}_1|\boldsymbol{x}_{t_i})$ is the solution to Eq. (9). If we assume $\phi_i$ solves Eq. (9) such that $q_{\phi_i}(\boldsymbol{x}_1|\boldsymbol{x}_{t_i}) = p(\boldsymbol{x}_1|\boldsymbol{x}_{t_i})$, Eq. (13) follows by simple calculation. The detailed proof can be found in the Appendix. □

In practice, by incorporating the equality constraint in Eq. (10) into the loss with a Lagrange multiplier, we obtain the UNSB objective for a single time-step $t_i$

$$\min_{\phi_i} \mathcal{L}_{\mathrm{UNSB}}(\phi_i, t_i) \coloneqq \mathcal{L}_{\mathrm{Adv}}(\phi_i, t_i) + \lambda_{\mathrm{SB}, t_i}\mathcal{L}_{\mathrm{SB}}(\phi_i, t_i). \tag{14}$$

Since it is impractical to use separate parameters $\phi_i$ for each time-step $t_i$ and to learn $p(\boldsymbol{x}_{t_{i+1}}|\boldsymbol{x}_{t_i})$ sequentially for $i = 0, \dots, N-1$, we replace $q_{\phi_i}(\boldsymbol{x}_1|\boldsymbol{x}_{t_i})$, which takes $\boldsymbol{x}_{t_i}$ as input, with a time-conditional DNN $q_\phi(\boldsymbol{x}_1|\boldsymbol{x}_{t_i})$, which shares a parameter $\phi$ for all time-steps $t_i$, and takes the tuple $(\boldsymbol{x}_{t_i}, t_i)$ as input. Then, we optimize the sum of $\mathcal{L}_{\mathrm{UNSB}}(\phi, t_i)$ over $i = 0, \dots, N-1$.

**Training.** For the training of our UNSB, we first randomly choose a time-step $t_i$ to optimize. To calculate $\mathcal{L}_{\mathrm{UNSB}}(\phi, t_i)$, we sample $\boldsymbol{x}_{t_i}$ and $\boldsymbol{x}_1 \sim \pi_1$, where $\pi_1$ denotes the target distribution. The sampling procedure of $\boldsymbol{x}_{t_i}$ will be described soon. The sample $\boldsymbol{x}_{t_i}$ is then passed through $q_\phi(\boldsymbol{x}_1|\boldsymbol{x}_{t_i})$ to obtain $\boldsymbol{x}_1(\boldsymbol{x}_{t_i})$ that refers to the estimated target data sample given $\boldsymbol{x}_{t_i}$. The pairs $(\boldsymbol{x}_{t_i}, \boldsymbol{x}_1(\boldsymbol{x}_{t_i}))$ and $(\boldsymbol{x}_1, \boldsymbol{x}_1(\boldsymbol{x}_{t_i}))$ are then used to compute $\mathcal{L}_{\mathrm{SB}}(\phi, t_i)$ and $\mathcal{L}_{\mathrm{Adv}}(\phi, t_i)$ in Eq. (9) and Eq. (10), respectively. Specifically, we estimate the entropy term in $\mathcal{L}_{\mathrm{SB}}$ with a mutual information estimator,

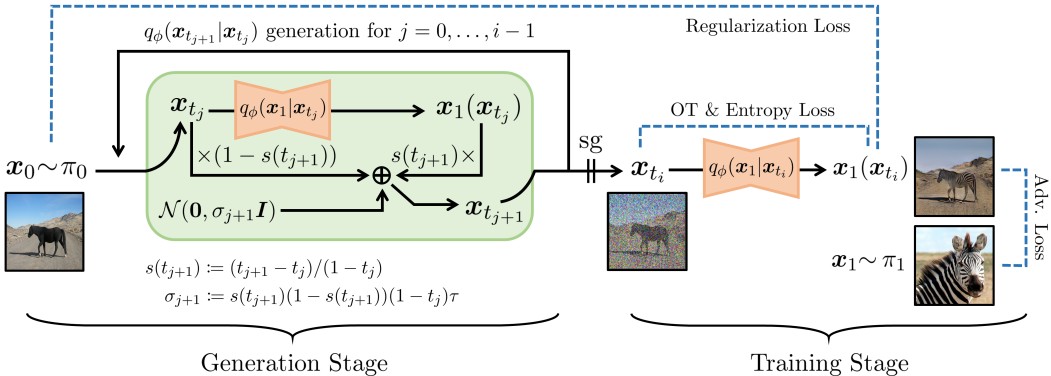

Figure 3: Generation and training process of UNSB for time step $t_i$. sg means stop gradient.

using the fact that for a random variable $X$, $I(X, X) = H(X)$ where $I$ denotes mutual information. We then estimate the divergence in $\mathcal{L}_{\text{Adv}}$ with adversarial learning. Intuitively, $\boldsymbol{x}_1$ and $\boldsymbol{x}_1(\boldsymbol{x}_{t_i})$ are "real" and "fake" inputs to the discriminator. This process is shown in the training stage of Figure 3.

**Generation of intermediate and final samples.** We now describe the sampling procedure for the intermediate samples for training and inference. We simulate the Markov chain Eq. (7) using $q_\phi$ as follows: given $\boldsymbol{x}_{t_j} \sim q_\phi(\boldsymbol{x}_{t_j})$ (note that $\boldsymbol{x}_{t_j} = \boldsymbol{x}_0 \sim \pi_0$ if $j = 0$), we predict the target domain image $\boldsymbol{x}_1(\boldsymbol{x}_{t_j}) \sim q_\phi(\boldsymbol{x}_1 | \boldsymbol{x}_{t_j})$. We then sample $\boldsymbol{x}_{t_{j+1}} \sim q_\phi(\boldsymbol{x}_{t_{j+1}})$ according to Eq. (11) by interpolating $\boldsymbol{x}_0$ and $\boldsymbol{x}_1(\boldsymbol{x}_{t_j})$ and adding Gaussian noise. Repeating this procedure for $j = 0 \ldots, i-1$, we get $\boldsymbol{x}_{t_i} \sim q_\phi(\boldsymbol{x}_{t_i})$. With optimal $\phi$, by Theorem 1, $\boldsymbol{x}_{t_i} \sim q_\phi(\boldsymbol{x}_{t_i}) = p(\boldsymbol{x}_{t_i})$. Thus, the trajectory $\{\boldsymbol{x}_1(\boldsymbol{x}_{t_i}) : i = 0, \ldots, N-1\}$ can be viewed as an iterative refinement of the predicted target domain sample. This process is illustrated within the generation stage of Figure 3.

### 4.1 COMBATING THE CURSE OF DIMENSIONALITY

**Advanced discriminator in adversarial learning.** One of the most important advantages of the formulation Eq. (14) is that we can replace the KL-divergence in $\mathcal{L}_{\text{Adv}}$ by any divergence or metric which measures discrepancy between two distributions. Such divergence or metric can be estimated through adversarial learning, i.e., its Kantorovich dual. For example,

$$2 \cdot D_{\text{JSD}}(q_{\phi_i}(\boldsymbol{x}_1) \| p(\boldsymbol{x}_1)) - \log(4) = \max_D \mathbb{E}_{p(\boldsymbol{x}_1)}[\log D(\boldsymbol{x}_1)] + \mathbb{E}_{q_{\phi_i}(\boldsymbol{x}_1)}[\log(1 - D(\boldsymbol{x}_1))] \quad (15)$$

where $D$ is a discriminator. This allows us to use various adversarial learning techniques to mitigate the curse of dimensionality. For instance, instead of using a standard discriminator which distinguishes generated and real samples on the instance level, we can use a Markovian discriminator which distinguishes samples on the patch level. The Markovian discriminator is effective at capturing high-frequency characteristics (e.g., style) of the target domain data (Isola et al., 2017).

**Regularization.** Furthermore, we augment the UNSB objective with regularization, which enforces the generator network $q_\phi$ to satisfy consistency between predicted $\boldsymbol{x}_1$ and the initial point $\boldsymbol{x}_0$:

$$\mathcal{L}_{\text{Reg}}(\phi, t_i) := \mathbb{E}_{p(\boldsymbol{x}_0, \boldsymbol{x}_{t_i})} \mathbb{E}_{q_\phi(\boldsymbol{x}_1 | \boldsymbol{x}_{t_i})}[\mathcal{R}(\boldsymbol{x}_0, \boldsymbol{x}_1)] \quad (16)$$

Here, $\mathcal{R}$ is a scalar-valued differentiable function which quantifies an application-specific measure of similarity between its inputs. In other words, $\mathcal{R}$ reflects our inductive bias for similarity between two images. Thus, the regularized UNSB objective for time $t_i$ is

$$\mathcal{L}_{\text{UNSB}}(\phi, t_i) := \mathcal{L}_{\text{Adv}}(\phi, t_i) + \lambda_{\text{SB}, t_i} \mathcal{L}_{\text{SB}}(\phi, t_i) + \lambda_{\text{Reg}, t_i} \mathcal{L}_{\text{Reg}}(\phi, t_i), \quad (17)$$

which is the final objective in our UNSB algorithm.

### 4.2 SANITY CHECK ON TOY DATA

**Two shells.** With the two shells data of Section 3, we show that representative SB methods suffer from the curse of dimensionality, whereas UNSB does not, given appropriate discriminator and regularization. For SB methods, we consider Sinkhorn-Knopp (SK), SB Conditional Flow Matching (SBCFM)

(Tong et al., 2023), Diffusion SB (DSB) (Bortoli et al., 2021), and SB-FBSDE (Chen et al., 2022). For baselines, we use recommended settings, and for UNSB, we train the discriminator to distinguish real and fake samples by input norms, and choose negative cosine similarity as $\mathcal{R}$ in Eq. (16).

Only $1k$ samples from each $\pi_0$, $\pi_1$ are used throughout the training. In Figure 4, we see all baselines either fail to transport samples to the target data manifold or fail to maintain high cosine similarity between input and outputs. On the other hand, UNSB is robust to dimension.

**Two Gaussians.** We also verify whether UNSB can learn the SB between two Gaussians. The SB between Gaussian distributions is also Gaussian, and its mean and covariance can be derived in a closed form (Bunne et al., 2023). Thus, we may measure the error between approximated and true solutions exactly. We consider learning the SB between $\mathcal{N}(-\mathbf{1}, \boldsymbol{I})$ and $\mathcal{N}(\mathbf{1}, \boldsymbol{I})$ in a 50-dimensional space. In Table 1, we see that UNSB recovers the mean and covariance of the ground-truth SB relatively accurately.

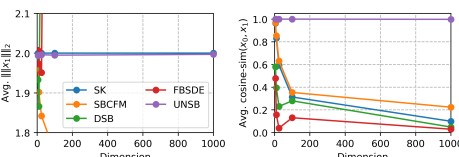

Figure 4: Results on two shells.

| Method | $\mu$ MSE | $\Sigma$ MSE |
|---|---|---|
| SK | 1.5e-5 | 3.1e-6 |
| SBCFM | 4.7e-5 | 2.3e-5 |
| DSB | 4.028 | 0.139 |
| SB-FBSDE | 2.974 | 2.5e-9 |
| UNSB | 0.008 | 6.4e-7 |

Table 1: Results on two Gaussians.

### 4.3 RELATION TO OTHER UNPAIRED TRANSLATION METHODS

**UNSB vs. GANs.** $N = 1$ version of UNSB is nearly equivalent to GAN-based translation methods such as CUT, i.e., UNSB may be interpreted as a multi-step generalization of GAN methods. Multi-step models are able to fit complex mappings that GANs are unable to by breaking down complex maps into a composition of simple maps. That is why diffusion models often achieve better performance than GANs (Salmona et al., 2022). Similarly, we believe the multi-step nature of UNSB arising from SB allow it to obtain better performance than GAN-based translation methods.

**UNSB vs. previous diffusion methods.** Previous diffusion-based methods such as SDEdit (Meng et al., 2022) also translate images along SDEs. However, the SDEs used by such methods (e.g., VP-SDE or VE-SDE) may be suboptimal in terms of transport cost, resulting in large NFEs. On the other hand, SB allows fast generation, as it learns the shortest path between domains. Furthermore, UNSB uses adversarial learning, so it may generate feasible samples even with NFE = 1. On the contrary, diffusion methods do not perform so well on complex tasks such as Horse2Zebra.

## 5 RESULTS FOR UNPAIRED IMAGE-TO-IMAGE TRANSLATION

In this section, we provide experimental results of UNSB for large scale unpaired image-to-image translation tasks.

**UNSB settings.** We use the Markovian discriminator in $\mathcal{L}_{\mathrm{Adv}}$ and the patch-wise contrastive matching loss (Park et al., 2020) in $\mathcal{L}_{\mathrm{Reg}}$. We set the number of timesteps as $N = 5$, so we discretize the unit interval into $t_0, t_1, \ldots, t_5$. We set $\lambda_{\mathrm{SB},t_i} = \lambda_{\mathrm{Reg},t_i} = 1$ and $\tau = 0.01$. We measure the sample quality of $\boldsymbol{x}_1(\boldsymbol{x}_{t_i})$ for each $i = 0, \ldots, N-1$. So, NFE = $i$ denotes sample evaluation or visualization of $\boldsymbol{x}_1(\boldsymbol{x}_{t_{i-1}})$. Other details are deferred to the Appendix.

**Evaluation.** We use four benchmark datasets: Horse2Zebra, Map2Cityscape, Summer2Winter, and Map2Satellite. All images are resized into $256 \times 256$. We use the FID score (Heusel et al., 2017) and the KID score (Chen et al., 2020) to measure sample quality.

**Baselines.** We note all SB methods SBCFM, DSB, and SB-FBSDE do not provide results on $\geq 256$ resolution images in their paper due to scalability issues. While we were able to make SBCFM work on 256 resolution images, the results were poor, so we deferred its results to the Appendix. Instead, we used Neural Optimal Transport (NOT) (Korotin et al., 2023) as a representative for OT. For GAN methods, we used CycleGAN (Zhu et al., 2017), MUNIT (Huang et al., 2018), DistanceGAN (Benaim & Wolf, 2017), GcGAN (Fu et al., 2019), and CUT (Park et al., 2020). For diffusion, we used SDEdit (Meng et al., 2022) and P2P (Hertz et al., 2022) with LDM (Rombach et al., 2022).

| Method | NFE | Time | Horse2Zebra | | Summer2Winter | | Label2Cityscape | | Map2Satellite | |
|---|---|---|---|---|---|---|---|---|---|---|
| | | | FID↓ | KID↓ | FID↓ | KID↓ | FID↓ | KID↓ | FID↓ | KID↓ |
| NOT | 1 | 0.006 | 104.3 | 5.012 | 185.5 | 8.732 | 221.3 | 19.76 | 224.9 | 16.59 |
| CycleGAN | 1 | 0.004 | 77.2 | 1.957 | 84.9 | 1.022 | 76.3 | 3.532 | 54.6 | 3.430 |
| MUNIT | 1 | 0.011 | 133.8 | 3.790 | 115.4 | 4.901 | 91.4 | 6.401 | 181.7 | 12.03 |
| Distance | 1 | 0.009 | 72.0 | 1.856 | 97.2 | 2.843 | 81.8 | 4.410 | 98.1 | 5.789 |
| GcGAN | 1 | 0.0027 | 86.7 | 2.051 | 97.5 | 2.755 | 105.2 | 6.824 | 79.4 | 5.153 |
| CUT | 1 | 0.0033 | 45.5 | 0.541 | 84.3 | 1.207 | 56.4 | 1.611 | 56.1 | 3.301 |
| SDEdit | 30 | 1.98 | 97.3 | 4.082 | 118.6 | 3.218 | – | – | – | – |
| P2P | 50 | 120 | 60.9 | 1.093 | 99.1 | 2.626 | – | – | – | – |
| Ours-best | 5 | 0.045 | **35.7** | 0.587 | **73.9** | **0.421** | 53.2 | 1.191 | 47.6 | 2.013 |

Table 2: NFE and generation time (sec.) per image and FID and KID×100 of generated samples.

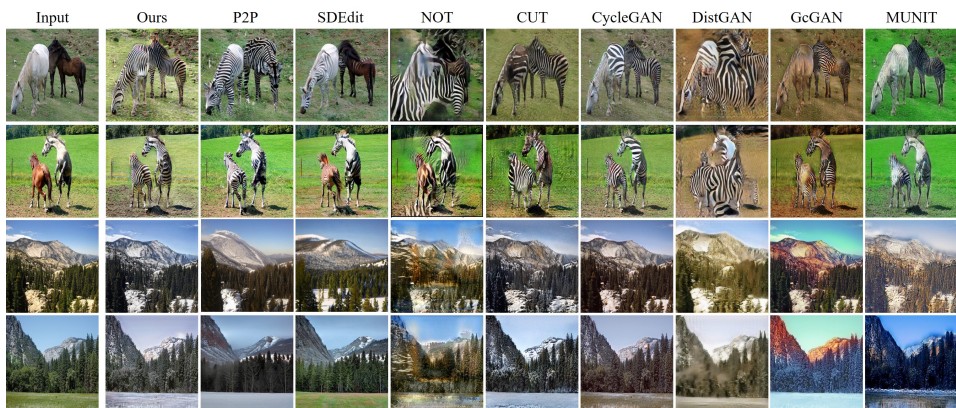

(a) Comparison on natural domain data.

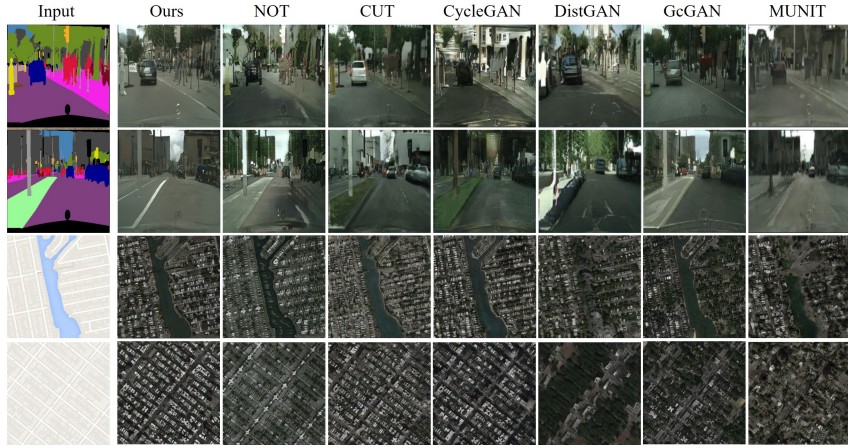

(b) Comparison on artificial domain data.

Figure 5: Qualitative comparison of image-to-image translation results from our UNSB and baseline I2I methods. Compared to other one-step baseline methods, our model generates more realistic domain-changed outputs while preserving the structural information of the source images.

Since LDM is trained on natural images, we can use LDM on Horse2Zebra and Summer2Winter, but not on Label2Cityscape and Map2Satellite. Hence, we omit diffusion results on those two tasks.

**Comparison results.** We show quantitative results in Table 2, where we observe our model outperforms baseline methods in all datasets. In particular, out model largely outperformed early GAN-based translation methods. When compared to recent models such as CUT, our model still shows better scores. NOT suffers from degraded performance in all of the datasets.

Qualitative comparison in Figure 5 provides insight into the superior performance of UNSB. Our model successfully translates source images to the target domain while preventing structural inconsis-

tency. For other GAN-based methods, the model outputs do not properly reflect the target domain information, and some models fail to preserve source image structure. For NOT, we observe that the model failed to overcome the curse of dimensionality. Specifically, NOT hallucinates structures not present in the source image (for instance, see the first row of Figure 5), leading to poor samples. For diffusion-based methods, we see they fail to fully reflect the target domain style.

**NFE analysis.** We investigate the relationship between NFE and the translated sample quality. In Figure 6 top, we observe that using NFE = 1 resulted in relatively poor performance, sometimes even worse than existing one-step GAN methods. However, we see that increasing the NFE consistently improved generation quality: best FIDs were achieved for NFE values between 3 and 5 for all datasets. This trend is also evident in our qualitative comparisons. Looking at Figure 1, we observe that as NFE increases, there are gradual improvements in image quality. However, UNSB at times suffers from the problem where artifacts occur with large NFE, as shown in Figure 6 bottom. We speculate this causes increasing FID for large NFEs on datasets such as Map2Satellite.

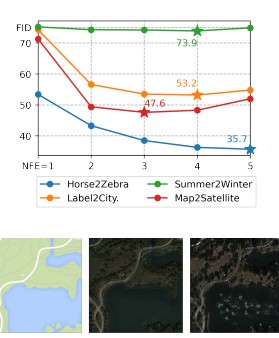

Figure 6: NFE analysis and failure case.

**Ablation study.** Unlike GAN-based methods, UNSB is able to generate samples in multiple steps. Also unlike SB methods, UNSB is able to use advanced discriminators and regularization. To understand the influence of those factors on the final performance, we performed an ablation study on the Horse2Zebra dataset. We report the results in Table 3. In the case where we use neither advanced discriminator nor regularization (corresponding to the previous SB method SBCFM), we see poor results. Then, as we add multi-step generation, advanced discriminator (Markovian discriminator, denoted "Patch"), and regularization, the result gradually improves until we obtain the best performance given by UNSB. This implies that the three components of UNSB play orthogonal roles.

| Disc. | Reg. | NFE | FID | KID |
|---|---|---|---|---|
| ✗ | ✗ | 5 | 230 | 18.8 |
| Instance | ✗ | 1 | 104 | 5.0 |
| Patch | ✗ | 1 | 66.3 | 1.6 |
| Patch | ✗ | 5 | 58.9 | 1.5 |
| Patch | ✓ | 5 | 35.7 | 0.587 |

Table 3: Ablation study.

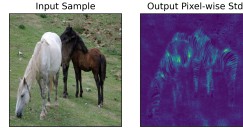

Figure 7: UNSB output variation.

**Stochasticity analysis.** SB is stochastic and should return diverse outputs. In Figure 7, we see meaningful variation in the generated images. Hence, UNSB indeed learns a stochastic map.

**Transport cost analysis.** SB solves OT, so inputs and outputs must be close. In Figure 8, we compare input-output distance for UNSB pairs and SK pairs computed between actual dataset images. we observe

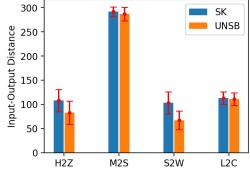

Figure 8: Input-output distance.

that input-output distance for UNSB is smaller than those of SK. This implies UNSB successfully generalizes beyond observed points to generate pairs which are close under $L_2$ norm and are realistic.

## 6 CONCLUSION

In this work, we proposed Unpaired Neural Schrödinger Bridge (UNSB) which solves the Schrödinger Bridge problem (SBP) via adversarial learning. UNSB formulation of SB allowed us to combine SB with GAN training techniques for unpaired image-to-image translation. We demonstrated the scalability and effectiveness of UNSB through various data-to-data or image-to-image translation tasks. In particular, while all previous methods for SB or OT fail, UNSB achieved results that often surpass those of one-step models. Overall, our work opens up a previously unexplored research direction for applying diffusion models to unpaired image translation.

## ETHICS AND REPRODUCIBILITY STATEMENTS

**Ethics statement.** UNSB extends diffusion to translation between two arbitrary distributions, allowing us to explore a wider range of applications. In particular, UNSB may be used in areas with beneficial impacts, such as medical image restoration. However, UNSB may also be used to create malicious content such as fake news, and this must be prevented through proper regulation.

**Reproducibility statement.** Pseudo-codes and hyper-parameters are described in the main paper and the Appendix.

## ACKNOWLEDGMENTS

This research was supported by the National Research Foundation of Korea (NRF) (RS-202300262527), Field-oriented Technology Development Project for Customs Administration funded by the Korean government (the Ministry of Science & ICT and the Korea Customs Service) through the National Research Foundation (NRF) of Korea under Grant NRF2021M3I1A1097910 & NRF2021M3I1A1097938, Korea Medical Device Development Fund grant funded by the Korea government (the Ministry of Science and ICT, the Ministry of Trade, Industry, and Energy, the Ministry of Health & Welfare, the Ministry of Food and Drug Safety) (Project Number: 1711137899, KMDF PR 20200901 0015), and Culture, Sports, and Tourism R&D Program through the Korea Creative Content Agency grant funded by the Ministry of Culture, Sports and Tourism in 2023.

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

# A    PROOFS

**Lemma 1** (Self-similarity). *Let $[t_a, t_b] \subseteq [0,1]$ and $\{\boldsymbol{x}_t\} \sim \mathbb{Q}^{\mathrm{SB}}$. The SB restricted to $[t_a, t_b]$, defined as the distribution of $\{\boldsymbol{x}_t\}|_{[t_a,t_b]} := \{\boldsymbol{x}_{t(s)} : s \in [0,1]\}$ where $t(s) := t_a + (t_b - t_a)s$ solves*

$$\min_{\mathbb{Q} \in \mathcal{P}(\Omega)} D_{\mathrm{KL}}(\mathbb{Q} \| \mathbb{W}^{\tau(t_b - t_a)}) \quad s.t. \quad \mathbb{Q}_0 = \mathbb{Q}_{t_a}^{\mathrm{SB}}, \ \mathbb{Q}_1 = \mathbb{Q}_{t_b}^{\mathrm{SB}}. \tag{18}$$

*Proof.* We claim $\boldsymbol{u}_t^{\mathrm{SB}}$ restricted to the interval $[t_a, t_b]$ is also a SB from $\mathbb{Q}_{t_a}^{\mathrm{SB}}$ to $\mathbb{Q}_{t_b}^{\mathrm{SB}}$. Suppose the claim is false, so there is another drift $\hat{\boldsymbol{u}}_t$ on $[t_a, t_b]$ such that

$$\mathbb{E}\left[\int_{t_a}^{t_b} \|\hat{\boldsymbol{u}}_t\|^2 \, dt\right] < \mathbb{E}\left[\int_{t_a}^{t_b} \|\boldsymbol{u}_t^{\mathrm{SB}}\|^2 \, dt\right] \quad \text{and} \quad \begin{cases} d\boldsymbol{x}_t = \hat{\boldsymbol{u}}_t \, dt + \sqrt{\tau} \, d\boldsymbol{w}_t, \\ \boldsymbol{x}_{t_a} \sim \mathbb{Q}_{t_a}^{\mathrm{SB}}, \ \boldsymbol{x}_{t_b} \sim \mathbb{Q}_{t_b}^{\mathrm{SB}}. \end{cases}$$

We can extend $\hat{\boldsymbol{u}}_t$ to the entire interval $[0,1]$ by defining

$$\hat{\boldsymbol{u}}_t = \begin{cases} \boldsymbol{u}_t^{\mathrm{SB}} & \text{if } 0 \le t < t_a, \\ \hat{\boldsymbol{u}}_t & \text{if } t_a \le t < t_b, \\ \boldsymbol{u}_t^{\mathrm{SB}} & \text{if } t_b \le t < 1. \end{cases}$$

We then have

$$\mathbb{E}\left[\int_0^1 \|\hat{\boldsymbol{u}}_t\|^2 \, dt\right] < \mathbb{E}\left[\int_0^1 \|\boldsymbol{u}_t^{\mathrm{SB}}\|^2 \, dt\right] \quad \text{and} \quad \begin{cases} d\boldsymbol{x}_t = \hat{\boldsymbol{u}}_t \, dt + \sqrt{\tau} \, d\boldsymbol{w}_t, \\ \boldsymbol{x}_0 \sim \pi_0, \ \boldsymbol{x}_1 \sim \pi_1, \end{cases}$$

which contradicts our assumption that $\boldsymbol{u}_t^{\mathrm{SB}}$ solves Eq. (3). We note that by change of time variable,

$$d\boldsymbol{x}_t = \boldsymbol{u}_t^{\mathrm{SB}} \, dt + \sqrt{\tau} \, d\boldsymbol{w}_t, \quad t_a \le t \le t_b$$

is equivalent to the SDE

$$d\boldsymbol{x}_s = (t_b - t_a)\boldsymbol{u}_{t(s)}^{\mathrm{SB}} \, ds + \sqrt{\tau(t_b - t_a)} \, d\boldsymbol{w}_s, \quad 0 \le s \le 1$$

By comparing the stochastic control formulation Eq. (3) with the original SBP Eq. (1), we see that this means the reference Wiener measure for the SBP restricted to $[t_a, t_b]$ has variance $\tau(t_b - t_a)$. $\quad\square$

**Lemma 2** (Static formulation of restricted SBs). *Let $t \in [t_a, t_b] \subseteq [0,1]$ and $\{\boldsymbol{x}_t\} \sim \mathbb{Q}^{\mathrm{SB}}$. Then*

$$p(\boldsymbol{x}_t | \boldsymbol{x}_{t_a}, \boldsymbol{x}_{t_b}) = \mathcal{N}(\boldsymbol{x}_t | s(t)\boldsymbol{x}_{t_b} + (1 - s(t))\boldsymbol{x}_{t_a}, s(t)(1 - s(t))\tau(t_b - t_a)\boldsymbol{I}) \tag{19}$$

*where $s(t) := (t - t_a)/(t_b - t_a)$ is the inverse function of $t(s)$. Moreover,*

$$\mathbb{Q}_{t_a t_b}^{\mathrm{SB}} = \underset{\gamma \in \Pi(\mathbb{Q}_{t_a}, \mathbb{Q}_{t_b})}{\arg\min} \mathbb{E}_{(\boldsymbol{x}_{t_a}, \boldsymbol{x}_{t_b}) \sim \gamma}[\|\boldsymbol{x}_{t_a} - \boldsymbol{x}_{t_b}\|^2] - 2\tau(t_b - t_a)H(\gamma). \tag{20}$$

*Proof.* Translate the SBP Eq. (18) into the static formulation, taking into account the reduced variance $\tau(t_b - t_a)$ of the reference Weiner measure $\mathbb{W}^{\tau(t_b - t_a)}$. $\quad\square$

*Proof of Theorem 1.* Let $t_a = t_i$ and $t_b = 1$ in Lemma 2. If the class of distributions expressed by $q_{\phi_i}(\boldsymbol{x}_1 | \boldsymbol{x}_{t_i})$ is sufficiently large, for any $\gamma \in \Pi(\mathbb{Q}_{t_i}, \mathbb{Q}_1)$,

$$d\gamma(\boldsymbol{x}_{t_i}, \boldsymbol{x}_1) = q_{\phi_i}(\boldsymbol{x}_{t_i}, \boldsymbol{x}_1) \tag{21}$$

for some $\phi_i$ which satisfies

$$D_{\mathrm{KL}}(q_{\phi_i}(\boldsymbol{x}_1) \| p(\boldsymbol{x}_1)) = 0. \tag{22}$$

Thus, Eq. (20) and Eq. (9) under the constraint Eq. (10) are equivalent optimization problems, which yield the same solutions, namely $p(\boldsymbol{x}_{t_i}, \boldsymbol{x}_1)$. Then, with optimal $\phi_i$,

$$q_{\phi_i}(\boldsymbol{x}_1 | \boldsymbol{x}_{t_i})p(\boldsymbol{x}_{t_i}) = q_{\phi_i}(\boldsymbol{x}_{t_i}, \boldsymbol{x}_1) = p(\boldsymbol{x}_{t_i}, \boldsymbol{x}_1). \tag{23}$$

Together with the observation that Eq. (11) is identical to Eq. (19), Eq. (23) implies Eq. (13). $\quad\square$

## B    OMITTED EXPERIMENT DETAILS

**Training.** All experiments are conducted on a single RTX3090 GPU. On each dataset, we train our UNSB network for 400 epochs with batch size 1 and Adam optimizer with $\beta_1 = 0.5$, $\beta_2 = 0.999$, and initial learning rate 0.0002. Learning rate is decayed linearly to zero until the end of training. All images are resized into $256 \times 256$ and normalized into range $[-1, 1]$. For SB training and simulation, we discretize the unit interval $[0, 1]$ into 5 intervals with uniform spacing. We used $\lambda_{\text{SB}} = \lambda_{\text{Reg}} = 1$ and $\tau = 0.01$. To estimate the entropy loss, we used mutual information neural estimation method Belghazi et al. (2018). To incorporate timestep embedding and stochastic conditioning into out UNSB network, we used positional embedding and AdaIN layers, respectively, following the implementation of DDGAN Xiao et al. (2022). For I2I tasks, we used CUT loss as regularization. On Summer2Winter translation task, we used a pre-trained VGG16 network as our feature selection source, following the strategy in the previous work by Zheng et al. (2021).

**Entropy approximation.** We first describe how to estimate the entropy of a general random variable. We observe that for a random variable $X$,

$$I(X, X) = H(X) - \underbrace{H(X|X)}_{=0} = H(X) \tag{24}$$

so mutual information may be used to estimate the entropy of a random variable. Works such as Belghazi et al. (2018) and Lim et al. (2020) provide methods to estimate mutual information of two random variables by optimizing a neural network. For instance, Belghazi et al. (2018) tells us that

$$I_\Theta(X, Z) := \sup_{\theta \in \Theta} \mathbb{E}_{\mathbb{P}_{XZ}}[T_\theta] - \log \left( \mathbb{E}_{\mathbb{P}_X \otimes \mathbb{P}_Z}[e^{T_\theta}] \right) \tag{25}$$

where $T_\theta$ is a neural network parametrized by $\theta \in \Theta$ is capable of approximating $I(X, Z)$ up to arbitrary accuracy. Hence, we can estimate $H(X) = I(X, X)$ by setting $X = Z$ and optimizing Eq. (25). In the UNSB objective Eq. (9), we use Eq. (25) to estimate the entropy of $X = Z = (\boldsymbol{x}_1, \boldsymbol{x}_{t_i})$ where $(\boldsymbol{x}_{t_i}, \boldsymbol{x}_1) \sim q_{\phi_i}(\boldsymbol{x}_{t_i}, \boldsymbol{x}_1)$ for $q_{\phi_i}$ defined in Eq. (8). In practice, we use the same neural net architecture for $T_\theta$ and the discriminator. We use the same network for $T_\theta$ for each time-step $t_i$ by incorporating time embedding into $T_\theta$. $T_\theta$ is updated once with gradient ascent every $q_{\phi_i}$ update, analogous to adversarial training.

**Evaluation.**

- Metric calculation codes
    - FID : https://github.com/mseitzer/pytorch-fid
    - KID : https://github.com/alpc91/NICE-GAN-pytorch
- Evaluation protocol: we follow standard procedure, as described in Park et al. (2020).
- KID measurement
    - DistanceGAN, MUNIT, NOT, SBCFM : trained from scratch using official code.
    - Baselines excluding above four methods : KID measured using samples generated by models from official repository.
- FID measurement
    - Horse2Zebra and Label2Cityscape : numbers taken from Table 1 in Park et al. (2020).
    - Summer2Winter and Map2Satellite : all baselines trained using official code.

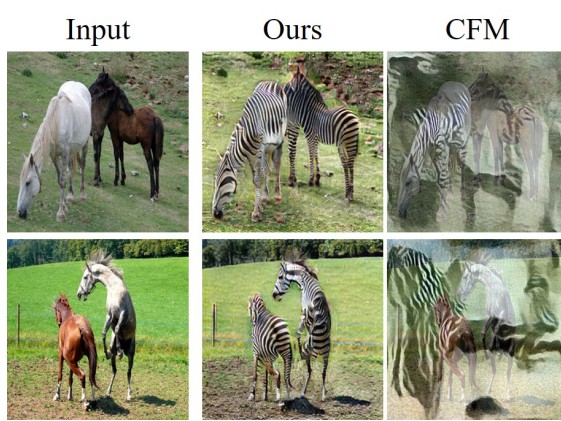

Figure 9: Visualization of our generated result and the result from conditional flow matching (CFM).

| Method | FID ↓ | KID↓ |
|--------|-------|------|
| SBCFM | 229.5 | 18.8 |
| Ours | **35.7** | **0.587** |

Table 4: Quantitative comparison result with another SB-based method on Horse2Zebra. The baseline model performance is severely degraded, while our generated results show superior perceptual quality.

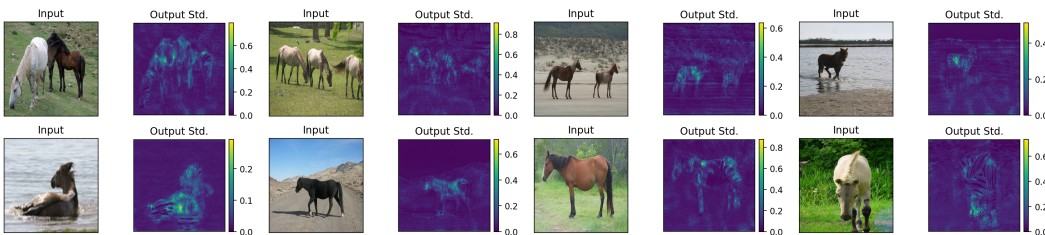

Figure 10: UNSB output pixel-wise standard deviation given input.

| Method | Time | NFE | FID↓ | KID↓ |
|--------|------|-----|------|------|
| EGSDE | 60 | 500 | 45.2 | 3.62 |
| StarGAN v2 | 0.006 | 1 | 48.55 | 3.20 |
| NOT | 0.006 | 1 | 51.5 | 3.17 |
| Ours | 0.045 | 5 | **37.87** | **1.54** |

Table 5: Male2Female results.

Figure 11: Male2Female samples.

## C  ADDITIONAL EXPERIMENTS

### C.1  OTHER SB METHODS

In this part, we compare our proposed UNSB with other Schrödinger bridge based image translation methods. The recent work of $I^2SB$ and InDI require supervised training setting, therefore we cannot compare our proposed method with the baselines. For most of other SB-based methods, they focus on relatively easy problem, where one side of distribution is simple. Since we firstly proposed applying SB-based approach to high-resolution I2I problem, there is no direct baseline to compare. However, we compared our model with SB-based approach of Schrödinger Bridge Conditional Flow Matching (SBCFM) (Shi et al., 2022) by adapting the SBCFM to high-resolution case.

In Table 4 and Fig. 9, we can see that the baseline SBCFM model could not generate the proper outputs where the outputs are totally unrelated to the input image. The results show that the model could not overcome the curse of dimensionality problem.

## C.2 MORE STOCHASTICITY ANALYSIS

In Figure 10, we visualize output pixel-wise standard deviation given input for more samples. We note that all pixel values are normalized into $[0, 1]$. A SB between two domains is stochastic, so it does one-to-many translation. Thus, a necessary condition for a model to be a SB is stochasticity. Figure 10 shows that UNSB satisfies this necessary condition.

Greater std for foreground pixels tells us UNSB does one-to-many generation, and generated images lie in the target domain. For instance, a model which simply adds Gaussian noise also does one-to-many generation, but is not meaningful. UNSB output shows high variation for locations which are relevant to the target domain (Zebra).

## C.3 MALE2FEMALE TRANSLATION

We provide results on the Male2Female translation task with the CelebA-HQ-256 dataset (Karras et al., 2018). Baseline methods are EGSDE (Zhao et al., 2022), StarGAN v2 (Choi et al., 2022), and NOT. UNSB is trained according to Appendix B. We used official pre-trained models for EGSDE and StarGAN v2 and trained NOT from scratch using the official code. We use the evaluation protocol in CUT (Park et al., 2020) to evaluate all models, i.e., we use test set images to measure FID. We note that (Zhao et al., 2022) and (Choi et al., 2022) use a different evaluation protocol in their papers, (they use train set images to measure FID) so the numbers in our paper and their papers may differ.

Results are shown in Table 5 and Figure 11. We note that UNSB produces outputs closer to the target image domain, as evidenced by FID and KID. Also, EGSDE, StarGAN v2, and NOT often fail to preserve input structure information. For instance, EGSDE fails to preserve eye direction or hair style. StarGAN v2 hallucinates earrings and fails to preserve hair style. NOT changes mouth shape (teeth are showing for outputs).

## C.4 ENTROPIC OT BENCHMARK

We evaluate UNSB on the entropic OT benchmark provided by Gushchin et al. (2023c) for $64 \times 64$ resolution images. We use $\epsilon = 0.1$ (using the notation in (Gushchin et al., 2023c)) and follow their training and evaluation protocol. Specifically, we trained UNSB to translate noisy CelebA images (i.e., $\boldsymbol{y}$-samples) to clean CelebA images (i.e., $\boldsymbol{x}$-samples). For comparison, we provide cFID numbers for SB $p(\boldsymbol{x}|\boldsymbol{y})$ learned by ENOT (Gushchin et al., 2023b) and unconditional $p(\boldsymbol{x}|\boldsymbol{y}) = p(\boldsymbol{x})$ where we use $5k$ ground-truth samples from $p(\boldsymbol{x})$. cFID for ENOT is taken from Table 7 in Gushchin et al. (2023c).

| Method | ENOT | Ours | $p(\boldsymbol{x})$ |
|--------|------|------|------|
| cFID | 40.5 | 63.0 | 104 |

Table 6: EOT benchmark.

In Table 6, we see that UNSB performs much better than the unconditional map $p(\boldsymbol{x}|\boldsymbol{y}) = p(\boldsymbol{x})$ but not quite as well as ENOT. We emphasize that this is only a preliminary result with ad-hoc hyper-parameter choices due to time constraints, and we expect better results with further tuning. Some points of improvement are:

- **Using more samples to measure cFID.** We only used 100 $\boldsymbol{x}$-samples for each $\boldsymbol{y}$-sample. The benchmark provides $5k$ $\boldsymbol{x}$-samples for each $\boldsymbol{y}$-sample, so using more $\boldsymbol{x}$-samples per $\boldsymbol{y}$-sample could improve the cFID.

- **Better architecture design.** UNSB architecture is specialized to $\geq 256 \times 256$ resolution images, whereas the benchmark consists of $64 \times 64$ resolution images. Currently, we set UNSB networks for this task as $\text{resize}(\boldsymbol{x}, 64) \circ G(\boldsymbol{x}) \circ \text{resize}(\boldsymbol{x}, 256)$, where $\text{resize}(\boldsymbol{x}, s)$ is a function which resizes images to $s \times s$ resolution, and $G(\boldsymbol{x})$ is the UNSB network used in our main experiments.

- **Better regularization.** CUT regularization is specialized to translation between clean images, whereas the benchmark consists of translation between noisy and clean images.

- **Longer training.** We only trained UNSB for $150k$ iterations. On, for instance, Male2Female, we needed to train about $1.7m$ iterations to get good results.

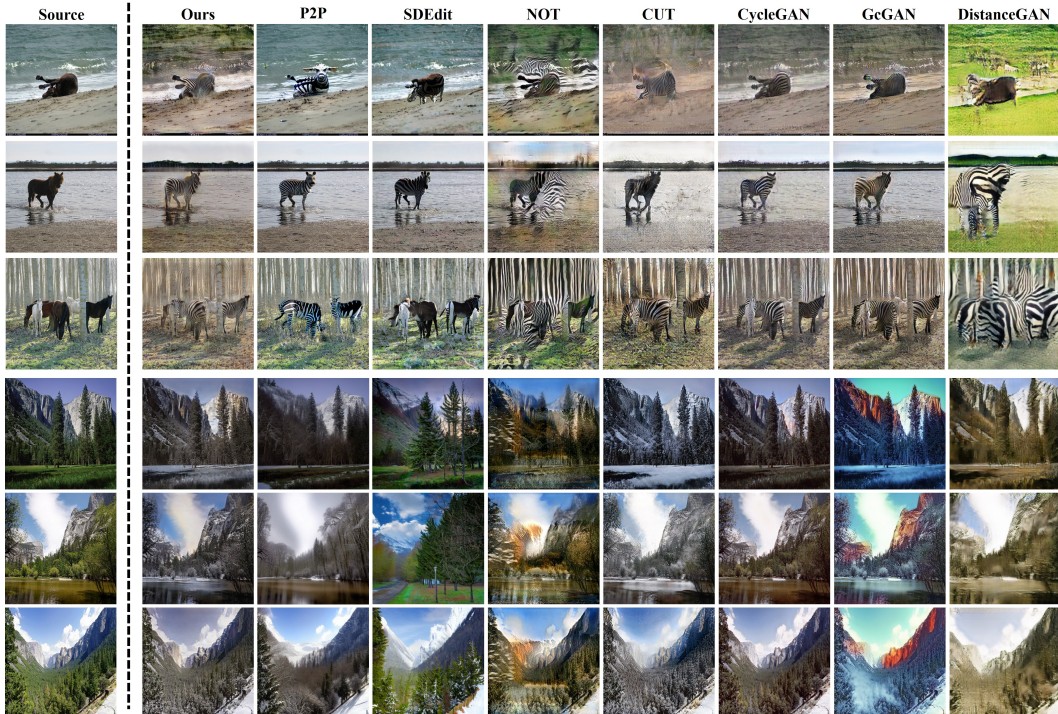

| Source | Ours | P2P | SDEdit | NOT | CUT | CycleGAN | GcGAN | DistanceGAN |

Figure 13: Additional samples on Horse2Zebra and Summer2Winter.

## C.5 ZEBRA2HORSE TRANSLATION

We show that UNSB is capable of reverse translation on tasks in our paper as well. Specifically, we compare CycleGAN and UNSB on Zebra2Horse translation task. In Table 7, we see that UNSB beats CycleGAN in terms of both FID and KID. In Figure 12, we observe UNSB successfully preserves input structure as well.

## C.6 MORE ABLATION STUDY

In Table 8, we provide another ablation study result on the Summer2Winter translation task, where we observe similar trends as before. Adding Markovian discrminator, regularization, and multi-step sampling monotonically increases the performance. This tells us the components of UNSB play orthogonal roles in unpaired image-to-image translation.

## C.7 ADDITIONAL SAMPLES

$256 \times 256$ **images.** In Fig. 13, 14, we show additional generated samples from our proposed UNSB and baseline models for our tasks in our main paper.

$512 \times 512$ **images.** In Fig. 15, we show samples from UNSB for Cat2Dog translation with AFHQ $512 \times 512$ resolution images, NFE = 5. This is a preliminary result with UNSB at training epoch 100.

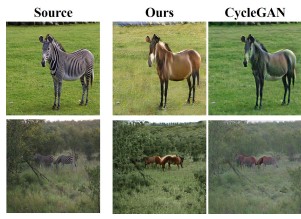

| Source | Ours | CycleGAN |

Figure 12: Zebra2Horse samples.

| Method | Time | NFE | FID | KID |
|---|---|---|---|---|
| CycleGAN | 0.004 | 1 | 135.75 | 2.998 |
| Ours | 0.045 | 5 | **134.09** | **2.581** |

Table 7: Zebra2Horse results.

| Disc. | Reg. | NFE | FID | KID |
|---|---|---|---|---|
| ✗ | ✗ | 5 | 254.2 | 23.91 |
| Instance | ✗ | 1 | 185.5 | 8.732 |
| Patch | ✗ | 1 | 155.5 | 7.697 |
| Patch | ✓ | 1 | 75.6 | 0.491 |
| Patch | ✓ | 5 | 73.9 | 0.421 |

Table 8: Summer2Winter ablation.

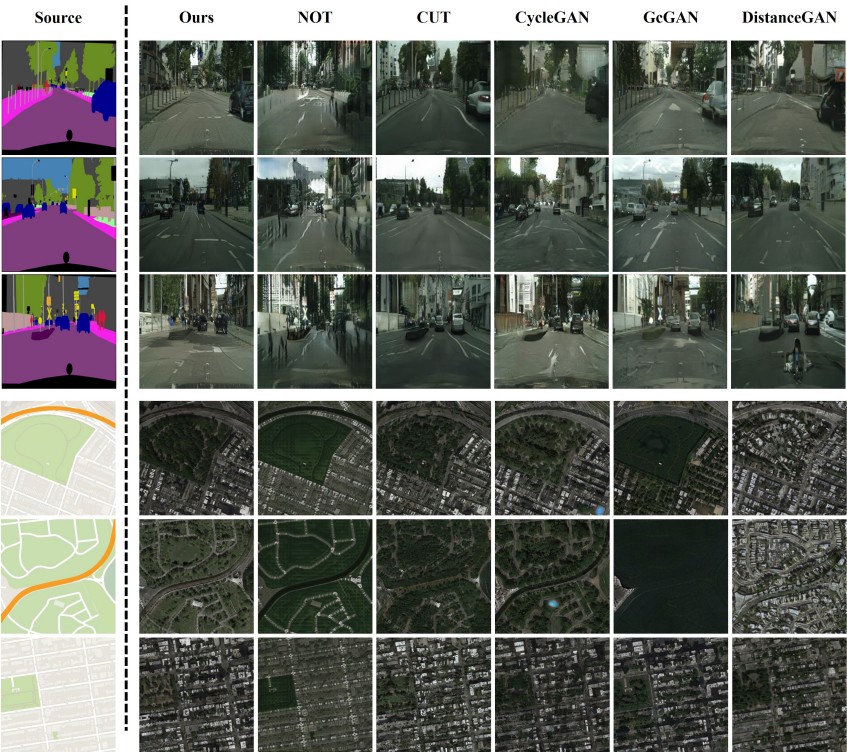

Figure 14: Additional samples on Label2Cityscapes and Map2Satellite dataset.

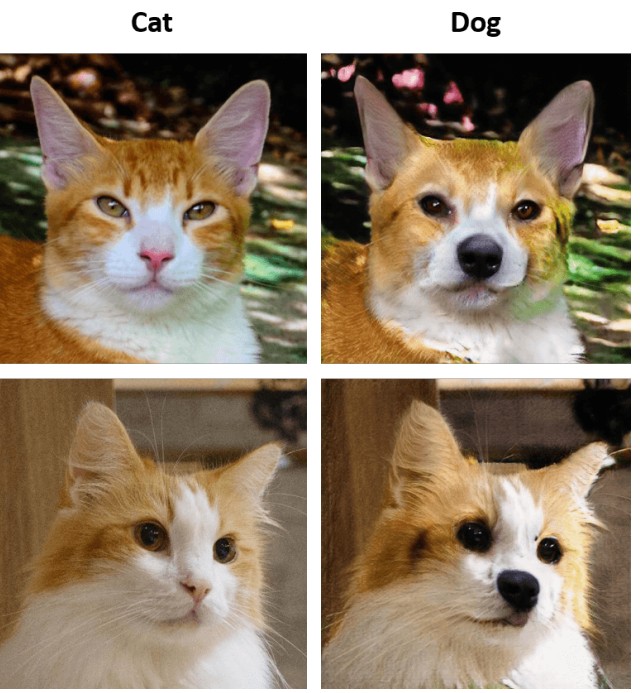

Figure 15: Cat2Dog on AFHQ $512 \times 512$ with UNSB. Preliminary result at training epoch 100.

## D   DISCUSSION ON SCENE-LEVEL VS. OBJECT-LEVEL TRANSFER

We note that unpaired image-to-image translation tasks can roughly be categorized into scene-level transfer and object-level transfer. In scene-level transfer (e.g., Horse2Zebra, Summer2Winter, etc.), mostly color and texture need to be altered during translation. In object-level transfer (e.g., Male2Female, Cat2Dog, etc.), object identities need to be altered during translation. We discuss differences between UNSB and diffusion-based translation methods on scene-level and object-level transfer tasks, respectively.

**Scene-level transfer.** While diffusion-based methods such as SDEdit are capable of translating between high-resolution images, we note that they do not perform so well on scene-level translation tasks considered in our paper. For instance, see Table 2 for SDEdit and P2P results on Horse2Zebra, where they show 97.3 FID and 60.9 FID, respectively. UNSB achieves 35.7 FID.

Figure 5a provides some clue as to why diffusion-based methods do not perform so well. P2P fails to preserve structural integrity (in the first row, the direction of the right horse is reversed) and SDEdit fails to properly translate images to zebra domain. Similar phenomenon occurs for Summer2Winter task as well. On the other hand, UNSB does not suffer from this problem as it uses a Markovian discriminator along with CUT regularization.

**Object-level transfer.** Diffusion-based methods perform reasonably on object-level translation tasks such as Male2Female or Wild2Cat. We speculate this is because object-level translation tasks use images with cropped and centered subjects, so it is relatively easy to preserve structural integrity. In fact, as shown in Figure 16, semantically meaningful pairs on such dataset pairs can be obtained even with Sinkhorn-Knopp (a computational optimal transport method), which does not rely on deep learning. However, Sinkhorn-Knopp is unable to find meaningful pairs on Horse2Zebra, where subjects are not cropped and centered.

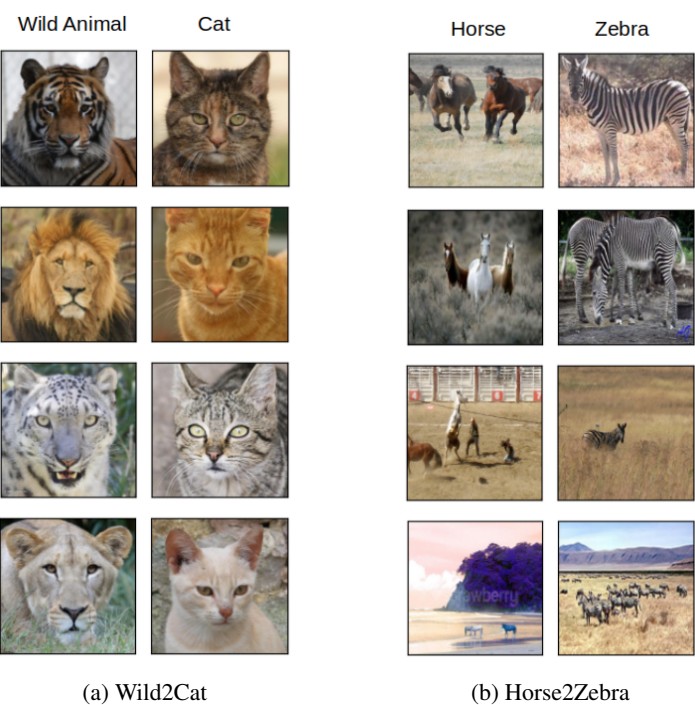

(a) Wild2Cat                              (b) Horse2Zebra

Figure 16: Unpaired image-to-image translation results with Sinkhorn-Knopp.

