# OpenReview forum: "Unpaired Image-to-Image Translation via Neural Schrödinger Bridge"
_ICLR.cc/2024/Conference — ICLR 2024 poster_

### Official Review · Reviewer_279L · 2023-10-12

**Soundness:** 3 good
**Presentation:** 3 good
**Contribution:** 3 good
**Rating:** 6
**Confidence:** 4

**Summary:**

The paper presents Unpaired Neural Schrödinger Bridge (UNSB), an innovative approach for unpaired image-to-image translation tasks, combining Schrödinger Bridge model with adversarial training. It expands the applicability of diffusion models by overcoming the Gaussian prior assumption limitation, which typically restricts them in unpaired image-to-image translations.The experiment results signal the potential of the proposed approach in dealing with high dimensional datasets, commonly referred to as the "curse of dimensionality." Experiments have shown the superiority of the proposed method where comparisons are conducted with their method and multiple baseline I2I methods on multiple I2I datasets. An ablation study is conducted with clear analyses.

**Strengths:**

1 The paper is aligned with recent advancements in stochastic generative modeling, image-to-image translations, and optimal transport, pushing the boundary of what's possible in these domains.

2 The experimentation shows promising results in I2I tasks in terms of multiple metrics.

3 The paper clearly explains the underlying mathematical conventions for Schrödinger Bridge and UNSB models and their connection with stochastic differential equations.

4 The authors have taken additional steps to ensure reproducibility of their work by providing code and detailed descriptions of hyperparameters in the paper.

**Weaknesses:**

There are no significant weaknesses. It could be valuable to include the reverse translations (such as Zebra to Horse, Winter to Summer) in the supplementary material, providing more comprehensive experimental results.

**Questions:**

See the weakness section.

---

> ### Author Response · Authors · 2023-11-16
> **Reply to Reviewer 279L**
>
> We thank the Reviewer for the insightful review, which has been incorporated into the revised paper. We address the Reviewer's concern below.
>
> > **Q1 : It could be valuable to include the reverse translations (such as Zebra to Horse, Winter to Summer) in the supplementary material, providing more comprehensive experimental results.**
>
> We kindly refer the Reviewer to Appendix C.5, where we provided an additional result on Zebra2Horse translation, and observed UNSB beats CycleGAN. We plan to include more results in the future.

---

> ### Comment · Reviewer_279L · 2023-11-18
> **Thanks for your response**
>
> The response adequately addressed my concerns, and I am inclined to keep my rating as 6.

---

> > ### Author Response · Authors · 2023-11-18
> >
> > Thank you for the positive score! Also, thank you for your time and the constructive review!

---

### Official Review · Reviewer_PUqf · 2023-10-28

**Soundness:** 3 good
**Presentation:** 3 good
**Contribution:** 2 fair
**Rating:** 6
**Confidence:** 3

**Summary:**

The paper proposes a method leveraging Schrödinger Bridge (SB) to bypass the limitation of applying Diffusion Models (which assume a Gaussian prior) to Image-to-Image (I2I) translation. Although previous approaches for utilizing SB to I2I translation have been explored, the proposed method, UNSB, is designed to be directly applied to I2I and scalable to high resolution images. The method is implemented as a composition of generators learned via adversarial learning that overcome the curse of dimensionality with advanced discriminators, showing success in several unpaired I2I translation tasks.

**Strengths:**

1. The formulation of SB as a composition of generators learned via adversarial learning seems novel. Moreover, the ability to apply patch-level discriminators is crucial for data with diverse textures as the ones considered in the paper.
2. The translation quality and faithfulness seem to be good.
3. The paper is well written and easy to follow.

**Weaknesses:**

1. I think that some relevant baselines were not included in the experimental section. Could the author compare the proposed method to EGSDE [1] (Diffusion-based model)?

[1] Zhao et al. EGSDE: Unpaired Image-to-Image Translation via Energy-Guided Stochastic Differential Equations. In NeurIPS, 2022.

2. The authors attribute the failure of SB for unpaired image-to-image translation to the curse of dimensionality. However, methods such as EGSDE and SDEdit do achieve successful translations in translating high resolution images. Could the authors elaborate on the difference between the cases?

3. Following my previous point, the datasets considered in the paper are all of “scene-level" image to image translation (in which mostly color and texture need to be altered during translation). In most of other I2I translation works, many "object-level" datasets are evaluated e.g. CelebA-HQ Male-to-Female, Cat-to-dog, etc. Do the authors expect different performance for the latter case? I believe it would be helpful to include benchmarks for at least some of these additional datasets to understand the strengths and the limitations of the proposed method.

**Questions:**

1. The description of the Stochasticity analysis (Fig. 7) is not clear to me. Could the author describe this ablation experiment and what does the generated map tell us? I understand that high pixel-wise std implies greater diversity, but why is it surprising that the std is greater for the foreground pixels? What is the scale of the std?

---

> ### Author Response · Authors · 2023-11-16
> **Reply to Reviewer PUqf**
>
> We thank the Reviewer for the insightful review, which has been incorporated into the revised paper. We address the Reviewer's concerns and questions below.
>
> > **Q1 : I think that some relevant baselines were not included in the experimental section. Could the author compare the proposed method to EGSDE [1] (Diffusion-based model)?**
>
> We kindly refer the Reviewer to Appendix C.3, where we compared UNSB with EGSDE, StarGAN v2, and NOT on the Male2Female translation task, and observed UNSB beats all baselines.
>
> > **Q2 : The authors attribute the failure of SB for unpaired image-to-image translation to the curse of dimensionality. However, methods such as EGSDE and SDEdit do achieve successful translations in translating high resolution images. Could the authors elaborate on the difference between the cases?**
>
> We kindly refer the Reviewer to Appendix D, where we added a discussion on the Reviewer’s question.
>
> > **Q3 : Following my previous point, the datasets considered in the paper are all of “scene-level" image to image translation (in which mostly color and texture need to be altered during translation). In most of other I2I translation works, many "object-level" datasets are evaluated e.g. CelebA-HQ Male-to-Female, Cat-to-dog, etc. Do the authors expect different performance for the latter case? I believe it would be helpful to include benchmarks for at least some of these additional datasets to understand the strengths and the limitations of the proposed method.**
>
> We kindly refer the Reviewer to Appendix C.3, where we performed an additional experiment on Male2Female, and observed good results for UNSB.
>
> > **Q4 : The description of the Stochasticity analysis (Fig. 7) is not clear to me. Could the author describe this ablation experiment and what does the generated map tell us? I understand that high pixel-wise std implies greater diversity, but why is it surprising that the std is greater for the foreground pixels? What is the scale of the std?**
>
> The following discussion has been added to Appendix C.2.
>
> Fig. 7 serves as evidence that UNSB indeed solves the SB problem. Specifically, a SB between two domains is stochastic, so it does one-to-many translation. Thus, a necessary condition for a model to be a SB is stochasticity. Fig. 7 shows that UNSB satisfies this necessary condition.
>
> Greater std for foreground pixels tells us UNSB does one-to-many generation, and generated images lie in the target domain. For instance, a model which simply adds Gaussian noise also does one-to-many generation, but is not meaningful. UNSB output shows high variation for locations which are relevant to the target domain (Zebra).
>
> Minimum and maximum values of pixel standard deviation in Figure 7 are 0.0031 and 0.7940, respectively. We note that all pixel values are normalized into [0,1].

---

> > ### Comment · Reviewer_PUqf · 2023-11-18
> >
> > I thank the authors for their detailed response.
> >
> > I find the results from the additional experiments on Male2Female to exhibit lower perceptual quality compared to the baselines (EGSDE and StarGAN-v2) even though it is not reflected in the FID for some reason. Similar degradation in the generated quality appear in the other experiments as well (e.g. blurred blending between the foreground zebras and the background). However, it indeed seems that the structural layout is better preserved using the proposed method, which is an appreciated contribution. I believe that a better balance of this tradeoff can be further explored in future work.
> >
> > I have updated my score accordingly.

---

> > > ### Author Response · Authors · 2023-11-18
> > >
> > > Thank you for raising the score! We also thank the Reviewer for proposing an interesting line of future research.

---

### Official Review · Reviewer_f3t2 · 2023-10-31

**Soundness:** 3 good
**Presentation:** 3 good
**Contribution:** 2 fair
**Rating:** 6
**Confidence:** 4

**Summary:**

The paper proposes a method for unpaired image-to-image translation based on Schrödinger bridge. The authors combine several ideas to obtain a practically-sounding method. At first, they propose to learn their model in a multistep manner, which is a natural and worth-considering given the recent success of diffusion models. Secondly, they propose to use the additional GAN loss to better match the target distribution. Additionally, the authors report the usefulness of the additional regularization applied at the training stage. Overall, the proposed approach demonstrates good practical results and has clear theoretical motivation.

**Update**: My concerns seems to be addressed. I rise my score to 6

**Strengths:**

The paper is overall-well written. It has clear motivation (unpaired image-to-image translation via entropic optimal transport (or Schrödinger bridge) problems. The idea of multistep approach, where the timeline of the Schrödinger process is discretized into several steps and a practitioner learn a kind of “denoising” model (an intuition from diffusion models) is fresh and worth to be considering. Moreover, it seems that the considering several steps indeed improves the qualitative performance, as shown by the practical experiments. The practical results are also encouraging.

**Weaknesses:**

My concerns are covered in the questions section (see below). I expect the authors to address them.

**Questions:**

- I have some doubts regarding the statement (end of page 1) that “no work has successfully trained SBs for direct translation between high-res images” and (end of page 2) “our work represents the first endeavor on this problem”. For example, [1], [2] deals with $64 \times 64$ images and tackles unpaired image-to-image problems. I am not sure, if the methods from [1], [2] works with $256 \times 256$, but, anyway, [1] and [2] should be covered as the related works and the competitive methods.
- Regarding the competitive methods (GANs), the authors miss the StarGAN and StarGAN v2 [3]. The latter model demonstrates a good quality on the unpaired image2image and I think this model is a good baseline method. Please, add it to the comparison.
- I have one concern about objective (14). The transition from constrained problem (9)-(10) to the combination of losses (14) is non-trivial and not fully theoretically justified. I am not sure, that the objective (14) indeed solves the SB problem. Some comments/clarifications will contribute to the clearness of the manuscript flow.
- I ask the authors to add the clarifications how to estimate the entropy with samples, a technique introduced at the beginning of page 6. It will facilitate the comprehension and smoothness of the reading.
- I strongly encourage the authors to evaluate how does their approach solve SB problem on a nontrivial setup beyond gaussian2gaussian case. In particular, there is a recent paper [4] which proposes benchmark pairs of distributions with known GT EOT/SB solutions, even for images data. The validation on such a benchmark will highly contribute to the quality of the paper and will probably answer the question (3).
- I have one more question regarding the baseline methods. The authors choose the work [5] (called them as NOT) “as the representative of OT [methods]”. This choice really surprised me because [5] also deals with SB problem, and, as I understand, their method is not designed for image data (at least, the authors of [5] does not consider image data use cases). That is why comparing with [5] seems strange. I recommend authors take [6, 7, 8] as the representative of OT methods instead. These methods also has a possibility to generate the samples of controllable diversity, have GAN-resembling objective. Moreover, they are designed specifically for unpaired image2image problems.

[1] Diffusion Schrödinger Bridge Matching, NeurIPS’2023

[2] Entropic Neural Optimal Transport via Diffusion Processes, NeurIPS’2023

[3] StarGAN v2: Diverse Image Synthesis for Multiple Domains, CVPR’2020

[4] Building the Bridge of Schrödinger: A Continuous Entropic Optimal Transport Benchmark, NeurIPS’2023

[5] Transport with support: Data-conditional diffusion bridges, arxiv

[6] Neural Optimal Transport, ICLR’2023

[7] Neural Monge Map estimation and its applications, TMLR

[8] Kernel Neural Optimal Transport, ICLR’2023

---

> ### Author Response · Authors · 2023-11-16
> **Reply to Reviewer f3t2**
>
> We thank the Reviewer for the insightful review, which has been incorporated into the revised paper. We address the Reviewer's concerns and questions below.
>
> > **Q1 : I have some doubts regarding the statement (end of page 1) that “no work has successfully trained SBs for direct translation between high-res images” and (end of page 2) “our work represents the first endeavor on this problem”. For example, [1], [2] deals with 64x64 images and tackles unpaired image-to-image problems. I am not sure, if the methods from [1], [2] works with 256x256, but, anyway, [1] and [2] should be covered as the related works and the competitive methods.**
>
> Thank you for informing us about [1] and [2]. We added a discussion regarding [1] and [2] in Section 2. We also attempted to implement [1,2] on our experiments, but faced scalability issues. Concretely, as noted in [1], [1] takes about 1 day to learn a SB between 28x28 grayscale images, with 2 GPUs. As noted in [2], [2] takes 7 days to learn a SB between 64x64 color images, with 2 GPUs. Compare this to UNSB: for instance, on Summer2Winter, UNSB takes 20 hours with 1 GPU to learn a SB between 256 x 256 color images.
>
> > **Q2 : Regarding the competitive methods (GANs), the authors miss the StarGAN and StarGAN v2 [3]. The latter model demonstrates a good quality on the unpaired image2image and I think this model is a good baseline method. Please, add it to the comparison.**
>
> We kindly refer the Reviewer to Appendix C.3, where we compared UNSB with StarGAN v2, EGSDE, and NOT on the Male2Female translation task, and observed UNSB beats all baselines.
>
> > **Q3 : I ask the authors to add the clarifications how to estimate the entropy with samples, a technique introduced at the beginning of page 6. It will facilitate the comprehension and smoothness of the reading.**
>
> Thank you for the constructive feedback. We provided more detail on entropy calculation in Appendix B of the revised paper.
>
> > **Q4 : I have one concern about objective (14). The transition from constrained problem (9)-(10) to the combination of losses (14) is non-trivial and not fully theoretically justified. I am not sure, that the objective (14) indeed solves the SB problem. Some comments/clarifications will contribute to the clearness of the manuscript flow. I strongly encourage the authors to evaluate how does their approach solve SB problem on a nontrivial setup beyond gaussian2gaussian case. In particular, there is a recent paper [4] which proposes benchmark pairs of distributions with known GT EOT/SB solutions, even for images data. The validation on such a benchmark will highly contribute to the quality of the paper and will probably answer this question.**
>
> Thanks for the important comment. The transition from constrained problem to an unconstrained formulation arises from using the Langragian and dual formulation. Furthermore, we would like to refer the Reviewer to Appendix C.4, where we added a result on the suggested benchmark [4], and observed positive results.
>
> > **Q5 : I have one more question regarding the baseline methods. The authors choose the work [5] (called them as NOT) “as the representative of OT [methods]”. This choice really surprised me because [5] also deals with SB problem, and, as I understand, their method is not designed for image data (at least, the authors of [5] does not consider image data use cases). That is why comparing with [5] seems strange. I recommend authors take [6, 7, 8] as the representative of OT methods instead. These methods also has a possibility to generate the samples of controllable diversity, have GAN-resembling objective. Moreover, they are designed specifically for unpaired image2image problems.**
>
> Thanks for pointing out the typo. We attached a wrong reference to NOT in our paper. NOT in our paper refers to [6] mentioned by the Reviewer, and all experiment results for NOT are carried out with [6]. We apologize for the mistake, and we fixed this reference in our revised paper.

---

> ### Author Response · Authors · 2023-11-21
> **Have we addressed your concerns?**
>
> Dear Reviewer f3t2,
>
> As the deadline for the Reviewer-Author discussion phase is fast approaching (there is only a day left), we respectfully ask whether we have addressed your questions and concerns adequately.
>
> Sincerely,
>
> The Authors.

---

> > ### Comment · Reviewer_f3t2 · 2023-11-23
> > **Response**
> >
> > Sorry for the delay. My concerns seems to be addressed. I rise my score to 6

---

> > > ### Author Response · Authors · 2023-11-23
> > >
> > > Thank you for raising the score! We thank the Reviewer for the constructive discussion, which we believe greatly improved our paper.

---

> ### Author Response · Authors · 2023-11-23
> **[Reminder] Summarization of our rebuttal**
>
> Dear reviewer f3t2,
>
> We would like to gently remind you that the **discussion period ends in approximately 10 hours**. We would appreciate it if you could let us know whether our comments addressed your concerns, as summarized below.
>
> - In **Section 2** we covered [1] and [2] as related works and competitive methods in the revised paper.
>
> - In **Section 5** we fixed a wrong reference for Neural Optimal Transport (NOT) [3], and clarified [3] is used as a representative for OT in all our experiments.
>
> - In **Appendix C.3** we conducted an additional experiment on Male2Female translation with CelebA-HQ-256 images with baselines EGSDE, StarGAN v2, and NOT, and observed UNSB beats the baselines.
>
> - In **Appendix B** we clarified how to approximate entropy in detail.
>
> - In **Appendix C.4** we validated UNSB on the suggested benchmark [4], and observed positive results.
>
> - In **Appendix C.7** we validated UNSB on Cat2Dog with AFHQ $512 \times 512$, which further demonstrates the scalability of our proposed approach.
>
> Best regards, Authors
>
> [1] Diffusion Schrödinger Bridge Matching, NeurIPS’2023
>
> [2] Entropic Neural Optimal Transport via Diffusion Processes, NeurIPS’2023
>
> [3] Neural Optimal Transport, ICLR’2023
>
> [4] Building the Bridge of Schrödinger: A Continuous Entropic Optimal Transport Benchmark, NeurIPS’2023

---

### Official Review · Reviewer_ek8v · 2023-11-02

**Soundness:** 3 good
**Presentation:** 3 good
**Contribution:** 3 good
**Rating:** 8
**Confidence:** 1

**Summary:**

This paper aims to address the problem that the diffusion model based unpaired image translation tasks rely on the Gaussion prior assumption. It resorts to Schrödinger Bridge and proposes the Unpaired Neural Schrödinger Bridge (UNSB) to express SB problem as a sequence of adversarial learning problems. By this formulation, SB can regarded as Lagrangian formulation under the constraint on the KL divergence between the true target distribution and the model distribution. This formulation leads to the composition of generators learned via adversarial learning that overcome the curse of dimensionality with advanced discriminators. Experiments demonstrated the effectiveness of the proposed method compared to its competitors.

**Strengths:**

The paper experimentally identifies the cause behind the failure of previous SB methods for image-to-image translation
as the curse of dimensionality, by a toy task.

The proposed method gives a new formulation of SB, which benefits on addressing the identified curse of dimensionality.

The proposed method shows better results than existing methods.

**Weaknesses:**

The paper claimed that "none of SB models so far have been successful at unpaired translation between highresolution images", however, the experiments in this paper do not include highresolution images yet (256x256 can not be regarded as highresolution). How will the proposed method perform on this setting is still unclear.

It is unclear that the findings by ablation study are still valid for other datasets?

When presenting the stochasticity analysis, it only gives results for one image. It would be better if the paper can provide statistical numbers across many different images/cases.

**Questions:**

no

---

> ### Author Response · Authors · 2023-11-16
> **Reply to Reviewer ek8v**
>
> We thank the Reviewer for the insightful review, which has been incorporated into the revised paper. We address the Reviewer's concerns and questions below.
>
> > **Q1 : The paper claimed that "none of SB models so far have been successful at unpaired translation between high resolution images", however, the experiments in this paper do not include high resolution images yet (256x256 cannot be regarded as high resolution). How will the proposed method perform on this setting is still unclear.**
>
> We kindly remind the Reviewer that multiple works [1,2,3] view 256x256 as high-resolution. Moreover, numerous representative methods for unpaired image-to-image translation [4,5] use 256x256 resolution images. Although we believe that our method can be scaled to higher resolution such as 512x512 or 1024x1024, and are running the experiments now, due to the limiting computational resource, it is a pity that we cannot provide the results during the rebuttal period.
>
> [1] High-Resolution Image Synthesis with Latent Diffusion Models, Rombach et al., CVPR, 2022.
>
> [2] EGSDE: Unpaired Image-to-Image Translation via Energy-Guided Stochastic Differential Equations, Zhao et al., NeurIPS, 2022.
>
> [3] Dual Diffusion Implicit Bridges for Image-to-Image translation, Su et al., ICLR, 2023.
>
> [4] Unpaired Image-to-Image Translation using Cycle-Consistent Adversarial Networks, Zhu et al., ICCV, 2017.
>
> [5] Contrastive Learning for Unpaired Image-to-Image Translation, Park et al., ECCV, 2020.
>
> > **Q2 : It is unclear that the findings by ablation study are still valid for other datasets?**
>
> We would like to refer the Reviewer to Appendix C.6, where we performed an additional ablation study on the Summer2Winter translation task, and observe similar trends as before.
>
> > **Q3 : When presenting the stochasticity analysis, it only gives results for one image. It would be better if the paper can provide statistical numbers across many different images/cases.**
>
> We kindly refer the Reviewer to Appendix C.2, where we provide more stochasticity analysis results.

---

> > ### Author Response · Authors · 2023-11-20
> > **Results on Higher Resolution Images**
> >
> > We gladly inform Reviewer ek8v that we obtained preliminary results on $512 \times 512$ resolution images. For more detail, we kindly refer the Reviewer to Figure 15 of our revised paper in Appendix C.7. This serves as evidence that UNSB works on even higher resolution images, and we believe this may address the Reviewer's first concern [**Q1**].

---

> ### Author Response · Authors · 2023-11-21
> **Have we addressed your concerns?**
>
> Dear Reviewer ek8v,
>
> As the deadline for Reviewer-Author discussion is fast approaching (there is only a day left), we respectfully ask whether we have addressed your questions and concerns adequately.
>
> Sincerely,
>
> The Authors.

---

> > ### Comment · Reviewer_ek8v · 2023-11-21
> >
> > Thanks the authors for the active responses. I will keep my initial score as my concerns have been adequately solved.

---

> > > ### Author Response · Authors · 2023-11-21
> > >
> > > Thank you for the discussion and the positive score!

---

### Author Response · Authors · 2023-11-16
**General Reply to All Reviewers**

We sincerely thank all Reviewers for the constructive feedback, which was incorporated into the revised version of our paper. Here, we summarize changes to our paper, such as additional experiments or clarifications. Modified contents have been highlighted red in the revised paper.

**Section 2.** We added missing references [1,2].

[1] Entropic neural optimal transport via diffusion processes, Gushchin et al., NeurIPS, 2023.

[2] Diffusion Schrödinger Bridge Matching, Shi et al., NeurIPS, 2023.

**Section 5.** We fixed a wrong reference for Neural Optimal Transport [3].

[3] Neural Optimal Transport, Korotin et al., ICLR, 2023.

**Appendix B.** We added more description on entropy approximation.

**Appendix C.2.** We added more results on stochasticity analysis.

**Appendix C.3.** We added a result on Male2Female translation, with baselines EGSDE, StarGAN v2, and NOT.

**Appendix C.4.** We added a result on the entropic OT benchmark [4].

[4] Building the bridge of Schrödinger: A continuous entropic optimal transport benchmark, Gushchin et al., NeurIPS Track on Datasets and Benchmarks, 2023.

**Appendix C.5.** We added a result on Zebra2Horse translation.

**Appendix C.6.** We added additional ablation study on Summer2Winter.

**Appendix D.** We discuss why diffusion-based methods perform well on some tasks (e.g., Male2Female, Cat2Dog), but not so well on others (e.g., Horse2Zebra, Summer2Winter).

---

### Author Response · Authors · 2023-11-20
**Result on $512 \times 512$ Resolution Images**

We would like to let the Reviewers know that we obtained encouraging results on even higher resolution images. In Figure 15 of our revised paper, we added Cat2Dog translation samples on AFHQ $512 \times 512$ resolution images. **We emphasize that this is a preliminary result with only 100 epochs of UNSB training, and we may obtain better results with longer training.**

---

### Meta-Review · Area_Chair_QBCy · 2023-12-05

**Metareview:**

This paper proposes a new method for unpaired image-to-image translation based on the Neural Schrödinger Bridge. Empirically, the proposed method is scalable and successfully solves various unpaired image-to-image translation tasks.

During the rebuttal, the authors successfully addressed the concerns of all reviewers by adding more experiments and baselines. All reviewers agreed that this paper should be accepted.

**Justification For Why Not Higher Score:**

Most of the reviewers think the overall contribution is incremental. Namely, they give a contribution rating of 2 and an overall rating of 6. Reviewer ek8v gave the highest score (8) yet had the lowest confidence (1).

**Justification For Why Not Lower Score:**

All reviewers agreed that this paper should be accepted.

---

### Decision · Program_Chairs · 2024-01-16

Accept (poster)